# Formation of highly oxygenated organic molecules from the oxidation of limonene by OH radical: significant contribution of H-abstraction pathway

Hao Luo[1,2], Luc Vereecken[3], Hongru Shen[1,2], Sungah Kang[3], Iida Pullinen[3a], Mattias Hallquist[4], Hendrik Fuchs[3,5], Andreas Wahner[3], Astrid Kiendler-Scharr[3†], Thomas F Mentel[3*], Defeng Zhao[1,2,6,7,8*]

[1]Department of Atmospheric and Oceanic Sciences & Institute of Atmospheric Sciences, Fudan University, Shanghai, 200438, China

[2]National Observations and Research Station for Wetland Ecosystems of the Yangtze Estuary, Fudan University, Shanghai, 200438, China

[3]Institute of Energy and Climate Research, IEK-8: Troposphere, Forschungszentrum Jülich, Jülich, 52425, Germany

[4]Department of Chemistry and Molecular biology, University of Gothenburg, Göteborg, 41258, Sweden

[5]Fachgruppe Physik, Universität zu Köln, Cologne, 50932, Germany

[6]Shanghai Frontiers Science Center of Atmosphere-Ocean Interaction, Fudan University, Shanghai 200438, China

[7]Institute of Eco-Chongming (IEC), 20 Cuiniao Rd., Chongming, Shanghai, 202162, China

[8]CMA-FDU Joint Laboratory of Marine Meteorology, Fudan University, Shanghai 200438, China

[a] Now at: Department of Applied Physics, University of Eastern Finland, Kuopio, 70210, Finland

[†]deceased on 6 February 2023

*Correspondence to*: Defeng Zhao (dfzhao@fudan.edu.cn), Thomas F Mentel (t.mentel@fz-juelich.de)

**Abstract.** Highly oxygenated organic molecules (HOM) play a pivotal role in the formation of secondary organic aerosol (SOA). Therefore, the distribution and yields of HOM are fundamental to understand their fate and chemical evolution in the atmosphere, and it is conducive to ultimately assess the impact of SOA on air quality and climate change. In this study, gas-phase HOM formed from the reaction of limonene with OH radical in photooxidation were investigated in the SAPHIR chamber (Simulation of Atmospheric PHotochemistry In a large Reaction chamber) using a time-of-flight chemical ionization mass spectrometer with nitrate reagent ion ($NO_3^-$-CIMS). A large number of HOM, including monomers ($C_{9-10}$) and dimers ($C_{17-20}$), were detected and classified into various families. Both closed-shell products and open-shell peroxy radicals ($RO_2$), were identified under low NO (0.06 - 0.1 ppb) and high NO conditions (17 ppb). $C_{10}$ monomers are the most abundant HOM products and account for over 80% total HOM. Closed-shell $C_{10}$ monomers were formed from two peroxy radical familie, $C_{10}H_{15}O_x\bullet(x=6\text{-}15)$ and $C_{10}H_{17}O_x\bullet(x=6\text{-}15)$, and their respective termination reactions with NO, $RO_2$, and $HO_2$. While $C_{10}H_{17}O_x\bullet$ is likely formed by OH addition to $C_{10}H_{16}$, the dominant initial step of limonene+OH, $C_{10}H_{15}O_x\bullet$, is likely formed via H-abstraction by OH. $C_{10}H_{15}O_x\bullet$ and related products contributed 41% and 42% of $C_{10}$-HOM at low and high NO, demonstrating that H-abstraction pathways play a significant role in HOM formation in the reaction of limonene+OH. Combining theoretical kinetic calculations, structure activity relationships (SARs), literature data, and the observed $RO_2$ intensities, we proposed tentative mechanisms of HOM formation from both pathways. We further estimated the molar yields of HOM to be $1.97^{+2.52}_{-1.06}$% and $0.29^{+0.38}_{-0.16}$% at low and high NO, respectively. Our study highlights the importance of H-abstraction by OH and provides yield and tentative pathways in the OH oxidation of limonene to simulate the HOM formation and assess their role in SOA formation.

## 1 Introduction

Biogenic volatile organic compounds (BVOC) are important precursors of atmospheric secondary organic aerosol particles (SOA) (Griffin et al., 1999; Kanakidou et al., 2005; Hallquist et al., 2009). In the Earth's atmosphere, monoterpenes ($C_{10}H_{16}$) are abundant BVOC with the second largest emission rate of approximately 150 Tg/year (Guenther et al., 2012). Among them, limonene accounts for 20% of total monoterpene emissions, ranking as the fourth largest (Guenther et al., 2012). In addition,

as an important ingredient of essential oil and common volatile chemical product (VCP), limonene is also widely used in house cleaning and personal care products due to its pleasant fragrance and antimicrobial property. Hence, limonene can play an important role in indoor SOA formation and indoor air quality (Waring, 2016; Rossignol et al., 2013; Nazaroff and Weschler, 2004). Although its concentration in the atmosphere is not the most abundant among monoterpenes, limonene has a high potential to form SOA due to its high SOA yield and high reactivity provided by the presence of an endocyclic and an exocyclic double-bond (Koch et al., 2000; Chen and Hopke, 2010; Gong et al., 2018; Saathoff et al., 2009).

Limonene can undergo gas-phase reactions rapidly with atmospheric oxidants such as the hydroxyl radical (OH) and ozone ($O_3$) in daytime and the nitrate radical ($NO_3$) and $O_3$ in nighttime forming peroxy radicals ($RO_2$) (Lee et al., 2006b; Eddingsaas et al., 2012; Vereecken and Peeters, 2012; Zhang et al., 2018; Yu et al., 1999; Vereecken and Peeters, 2000). In the reaction of limonene with OH, the main atmospheric daytime loss process of limonene, OH addition to one of the olefinic carbon atoms is the dominant channel, although abstraction of an allylic hydrogen atom also occurs (Rio et al., 2010; Jokinen et al., 2014). Via OH addition, the reaction forms $C_{10}H_{17}O_3\bullet$ as a first-generation $RO_2$, which under oxidative conditions further react with $RO_2$, $HO_2$, and NO forming oxidation products including limonaldehyde ($C_{10}H_{16}O_2$), limonaketone ($C_9H_{14}O$), limonic acid ($C_{10}H_{16}O_3$), ketolimononic acid ($C_9H_{14}O_4$), peroxo-pinic acid or its isomer ($C_{10}H_{14}O_5$), nitric acid ester ($C_9H_{13}NO_7$), some organic acids with low molecular weight (formic acid, acetic acid, butyric acid, methacrylic), and formaldehyde (HCHO) (Pang et al., 2022; Friedman and Farmer, 2018; Romonosky et al., 2015; Jaoui et al., 2006; Lee et al., 2006a; Hakola et al., 1994). Although earlier studies have gained valuable insights into the products and mechanism of the reaction with OH, many products and their formation mechanisms are not well elucidated. For example, highly oxygenated organic molecules (HOM) products and their formation mechanism remain elusive.

HOM was first discovered to be produced in forests and is defined as compounds containing six or more oxygen atoms formed in the gas phase via autoxidation (Bianchi et al., 2019; Ehn et al., 2012; Ehn et al., 2014). Among the myriad of oxygenated products, HOM was found of great importance for the mass of SOA, new particle formation and particle growth (Jokinen et al., 2015; Jokinen et al., 2014; Ehn et al., 2014; Mentel et al., 2015; Molteni et al., 2019; Kirkby et al., 2016; McFiggans et al., 2019; Crounse et al., 2013; Kenseth et al., 2018; Ehn et al., 2017; Krechmer et al., 2015; Quéléver et al., 2019; Tröstl et al., 2016). Typically, HOM are generated from the autoxidation of peroxy radicals ($RO_2$), where a new, more oxygenated $RO_2$ can be formed after an intramolecular H-shift followed by $O_2$ addition in the resulting alkyl radical (Bianchi et al., 2019; Vereecken and Nozière, 2020; Crounse et al., 2013; Møller et al., 2019; Vereecken et al., 2007; Ehn et al., 2017; Nozière and Vereecken, 2019; Ehn et al., 2014). Bimolecular termination of the autoxidation chain occurs when an $RO_2$ intermediate react with $HO_2$, $RO_2$, and NO to form a series of products (see below), which can be classified as organic hydroperoxide (ROOH, R1a), alcohol (ROH, R2), and carbonyl (R=O, R2), alkoxy radicals (RO, R3-4), or organic nitrates (ON, R5). The molecular masses of termination products are $M+1$, $M-15$, $M-17$ and $M+30$, respectively, for an $RO_2$ with molecular mass of $M$ (Mentel et al., 2015; Ehn et al., 2014). Meanwhile, unimolecular termination pathways generally lead to the formation of carbonyls (R=O, R6). In addition, two $RO_2$ can also undergo an accretion reaction to form HOM dimers (R7) as the reactions shown below (Berndt et al., 2018a; Valiev et al., 2019; Zhao et al., 2018b). Note that $RO_2$ reaction in R2 is considered for primary and secondary $RO_2$. For tertiary $RO_2$s, carbonyls cannot be formed. In addition, the unimolecular isomerization of RO (from R1b and R3-4) could produce $RO_2$ in the same $RO_2$ family.

$$RO_2 + HO_2 \rightarrow ROOH + O_2 \qquad (R1a)$$
$$\rightarrow RO + OH \qquad (R1b)$$
$$RO_2 + R'O_2 \rightarrow R=O + R'OH \qquad (R2)$$
$$RO_2 + R'O_2 \rightarrow RO + R'O + O_2 \qquad (R3)$$
$$RO_2 + NO_3 \rightarrow RO + NO_2 + O_2 \qquad (R4)$$
$$RO_2 + NO \rightarrow RONO_2 + O_2 \qquad (R5)$$
$$RO_2 \xrightarrow{H-shift} R=O + OH \qquad (R6)$$

$$RO_2+R'O_2 \rightarrow ROOR'+O_2 \qquad (R7)$$

HOM formation in OH-initiated oxidation of monoterpenes such as α-pinene and β-pinene has been studied in the laboratory (Berndt, 2021; Xu et al., 2019; Berndt et al., 2016; Kirkby et al., 2016; Ehn et al., 2014; Shen et al., 2022), and is assumed to mainly start with the OH addition to C=C double bond. The alkyl radical adduct can rapidly add $O_2$ to the radical site to form a peroxy radical (Finlayson-Pitts and Pitts, 2000; Ziemann and Atkinson, 2012). A number of monomers and dimers products from several monoterpenes following the reaction with hydroxyl radical (OH) were observed, such as $C_{10}H_{14-18}O_{7-11}$ and $C_{19-20}H_{28-32}O_{10-18}$ (Berndt et al., 2018a; Shen et al., 2022; Xu et al., 2019). Despite the molecular similarities of HOM monomers or dimers products from different monoterpenes (Jokinen et al., 2015; Ehn et al., 2014; Ehn et al., 2012), HOM yields are quite different between different monoterpenes in a given oxidation reaction (e.g. with $O_3$ or OH radical) (Jokinen et al., 2015). It follows that the HOM formation mechanisms for other monoterpenes may not necessarily be applicable to limonene. Although HOM formation from limonene by ozonolysis and $NO_3$ has been reported (Jokinen et al., 2015; Guo et al., 2022; Mayorga et al., 2022), to the best of our knowledge, no studies have systematically reported HOM distribution in limonene+OH. In the ozonolysis of limonene, Jokinen et al. (2014) attributed some HOM ($C_{10}H_{17}O_x \bullet_{(x=5,7,9)}$ ($RO_2$)) to the reaction of limonene with OH produced in ozonolysis by comparing the mass spectra in the presence or absence of an OH scavenger. The composition and the formation mechanism of HOM in limonene photooxidation by OH remains unclear. Moreover, although the HOM yield of limonene+OH was determined, this yield was estimated based on the difference of the HOM yield in the ozonolysis with and without OH scavenger (HOM yields: 5.3% (limonene+$O_3$); 0.93% (limonene+OH)) (Jokinen et al., 2015). It is necessary to confirm the yield by its direct determination in the reaction of limonene+OH. Interestingly, in our previous study, we found that hydrogen abstraction is important for HOM formation in the reaction of α-pinene+OH (Shen et al., 2022), where intermediate alkoxy radicals oxidation steps aid autoxidation by breaking the six-membered and four-membered rings. As limonene+OH is known to have a high H-abstraction contribution of ~34% (Rio et al., 2010; Dash and Rajakumar, 2015; Braure et al., 2014), it is interesting to compare these two systems due to the similar but subtly different structures of α-pinene and limonene, with both containing a six-membered ring whereas limonene does not contain a four-membered ring.

In this study, we investigated HOM from the photooxidation of limonene with OH at low NO (0.06 - 0.1 ppb) and high NO (17 ppb) in the SAPHIR chamber at Forschungszentrum Jülich. HOM were classified into monomers and dimers, and the product distribution is reported. The yields of HOM from limonene with OH were estimated. Formation mechanism of HOM in the OH reaction is proposed based on molecular formula of HOM and quantum chemical calculation as well as structure activity relationships (SARs) of $RO_2$ autoxidation rates. The relative importance of OH addition to C=C double bond and hydrogen abstraction by OH is discussed. We further investigated the effects of $NO_x$, which changes HOM composition via altering the $RO_2$ fate.

## 2 Methodology

### 2.1 Experiment design and set-up

The experiments were conducted in the "SAPHIR" chamber (Simulation of Atmospheric PHotochemistry In a large Reaction chamber) at Forschungszentrum Jülich, Germany. The details of the chamber have been described in the previous studies (Zhao et al., 2018a; Zhao et al., 2015b; Zhao et al., 2015a; Zhao et al., 2021; Rohrer et al., 2005; Shen et al., 2021; Guo et al., 2022). In brief, SAPHIR is a 270 $m^3$ Teflon chamber equipped with a louvre system to switch between natural sunlight for illumination and dark conditions. In this study, the experiments were conducted in sun light with the louvres opened. To avoid possible interference due to long reaction time, the subsequent discussion focuses on the early stage (15 min) of the experiment. The initial experimental conditions are shown in Table S1.

Gas and particle phase species were characterized by a comprehensive set of instruments with the details described before (Zhao et al., 2015b). A Proton Transfer Reaction Time-of-Flight Mass Spectrometer (PTR-ToF-MS, Ionicon Analytik, Austria)

was used for measuring VOC. A $NO_x$ analyzer (ECO PHYSICS TR480) and an UV photometer $O_3$ analyzer (ANSYCO, model O341M) were used to measure the concentrations of $NO_2$, NO and $O_3$, respectively. A laser induced fluorescence system (LIF) was used to measure the concentrations of OH, $HO_2$ and $RO_2$ (Fuchs et al., 2012). Note that the potential artefact in $HO_2$ measurements from the concurrent chemical conversion of $RO_2$ in instrument making use of chemical conversion of $HO_2$ by the reaction with NO can be avoided in this study through NO used, so that no corrections of $HO_2$ concentration measurements are required. The detection of $RO_2$ radicals relies on the conversion of $RO_2$ to $HO_2$ in their reactions with NO. We applied a correction to $RO_2$ concentrations (Fuchs et al., 2011). We would like to note that only a few nitrated $RO_2$ were observed to not form $HO_2$ in the reaction with NO. In this study, we do not expect that there were a large contribution of nitrate $RO_2$ to the sum of all $RO_2$ as in photochemistry experiments, as there is no significant fraction of nitrate $RO_2$ formed. A Scanning Mobility Particle Sizer Spectrometer (SMPS, TSI, DMA3081/CPC3785) was used to obtain particle number distributions. Temperature and relative humidity were continuously measured.

Before an experiment was conducted, the chamber was flushed with high purity synthetic air (purity>99.9999% $N_2$ and $O_2$). Experiments were conducted at ~75% RH initially. In low NO condition, no NO was added and the background concentration of NO, which mainly stems from HONO photolysis produced via a photolytic process from Teflon wall, is 0.06 - 0.1 ppb (Fig. S1). OH radicals were generated from the photolysis of HONO in both low and high NO experiments and the HONO was formed from the Teflon chamber wall via a photolytic process. The details have been described by Rohrer et al. (2005). $HO_2$ was produced from the reaction of $O_2$ with RO, which can be formed in the reaction of $RO_2$+NO in photo-oxidation during the experiments. The concentration of limonene was 7 ppb. The reaction time after the roof opened was 8 hours. At high NO condition, 17 ppb NO was added into the chamber first and limonene was sequentially added after half an hour. The louvres were opened ~40 min after adding limonene. To estimate the impact of ozone oxidation during photoxidation, we calculated the reaction rates of VOC+ OH and VOC+$O_3$ in the experiments of this study (Fig. S2). During the first 15 min, VOC+OH accounted for >99% limonene loss in both low and high NO condition. We would like to note that the low NO does not refer to the case where $RO_2$ loss is dominated by the reaction with $HO_2$, e.g., in remote ocean environment. At low NO the $RO_2$ loss was estimated to be dominated by its reactions with NO in the early period (within ~15min, $RO_2$ + $HO_2$ contributed ~15% of $RO_2$ loss) and in later periods a significant fraction of $RO_2$ loss was also contributed by the reaction of $RO_2$ with $HO_2$ (Fig. S3), based on the measured $NO_x$, $RO_2$, and $HO_2$ concentrations and their rate constants for the reactions with $RO_2$ (Jenkin et al., 1997; Jenkin et al., 2019). At high NO, the $RO_2$ fate was by far dominated by the $RO_2$+NO reaction.

## 2.2 Characterization of HOM

HOM were characterized by a Chemical Ionization time-of-flight Mass Spectrometer (CIMS, Aerodyne Research Inc., USA) with nitrate ($^{15}NO_3^-$) as the reagent ion, which has a mass resolution of ~4000 (m/dm). The details of the instrument are described in previous publications (Pullinen et al., 2020; Mentel et al., 2015; Ehn et al., 2014). Briefly, $^{15}NO_3^-$ produced from $^{15}N$ nitric acid, was used as the reagent ion to distinguish complexation with the reagent ion from $NO_3$ groups in target molecules. $NO_3^-$-CIMS is suitable for detecting oxygenated organic compounds with high oxygen number. The mass spectra were analysed by Tofware (version 2.5.7, Tofwerk/Aerodyne) in Igor Pro (version 6.37 WaveMetrics, Inc.). In this study, the mass spectra of HOM products during the first 15 min after louvres opening were analysed because the particle number concentration (<30 #/$cm^{-1}$) remained low in the initial phase of the reaction. After attributing molecular formulas of HOM to different m/z, their concentrations were calculated using the calibration coefficient of $H_2SO_4$ (C: $2.5×10^{10}$ molecule·$cm^{-3}$·$nc^{-1}$) as described before (Zhao et al., 2021; Pullinen et al., 2020), where the charge efficiency of HOM and $H_2SO_4$ was assumed to be close to the collision limit (Pullinen et al., 2020; Ehn et al., 2014). The details of the calibration with $H_2SO_4$ were described in Supplement Sect. S1. The loss of HOM was corrected by using a wall loss rate of $2.2×10^{-3}$ $s^{-1}$ in fan-on condition and ~$6.0×10^{-4}$ $s^{-1}$ in fan-off condition as quantified previously (Guo et al., 2022; Zhao et al., 2018a), and a dilution loss rate ~$1×10^{-6}$ $s^{-1}$ (Zhao et al., 2015b). For details, we refer to these latter publications; briefly, the wall loss rate of HOM in our chamber

was estimated as that of the decay of organic vapor (such as $C_{10}H_{15}NO_{9-12}$ (nitrated compounds) and $C_{10}H_{14}O_{8-11}$ (non-nitrated compounds) in the reaction of limonene with OH in the presence of NO) concentrations in the dark (Guo et al., 2022). Overall, wall loss correction and dilution correction only affect the HOM yield by ~5.8% and <1%, respectively.

**2.3 Data analysis**

The HOM yield was obtained using the concentration of the HOM, divided by the concentration of limonene consumed by OH, which is the dominant oxidant of limonene and accounts for over 99% of limonene loss rate. HOM yield was calculated over the first 15 min after louvres opening as followed:

$$Y = \frac{[HOM]}{\triangle[VOC]_r} = \frac{I_{HOM}*C}{\triangle[VOC]_r} \qquad (Eq. 1)$$

where [*HOM*] means the concentrations of total HOM corrected for wall loss and dilution loss, $\triangle[VOC]_r$ is the consumption concentrations of limonene corrected for wall loss and dilution loss, $I_{HOM}$ is the total signal intensity of HOM normalized to the total signal, and *C* is the calibration coefficient of $H_2SO_4$. The calibration and wall loss and dilution loss correction are described in detail in the Supplement (S1).

Based on the $C_{10}$ closed-shell products from unimolecular and bimolecular reactions of the $C_{10}$ peroxy radical families $C_{10}H_{15}O_x\bullet$ and $C_{10}H_{17}O_x\bullet$, $C_{10}H_{16}O_x$ can be divided into carbonyls (R=O) and epoxides from $C_{10}H_{17}O_x\bullet$, as well as alcohols (ROH) and hydroperoxides (ROOH) from $C_{10}H_{15}O_x\bullet$. The contribution of $C_{10}H_{17}O_x\bullet$-related products to $C_{10}H_{16}O_x$, was quantified as follows. For a HOM-RO$_2$, the production rate of alcohols (ROH) can be obtained according to R2.

$$\frac{d[ROH]}{dt} = \alpha k_{RO2+RO2}[RO_2][RO_2]^T \qquad (Eq.2)$$

The production rate of hydroperoxides (ROOH) can be obtained according to R1a, which forms ROOH with a yield β, where β is close to 1 for most RO$_2$ (Jenkin et al., 2019).

$$\frac{d[ROOH]}{dt} = k_{RO2+HO2}[RO_2][HO_2]\beta \qquad (Eq.3)$$

The production rate of carbonyls (R=O) can be obtained according to R2 and R6.

$$\frac{d[R=O]}{dt} = (1-\alpha)k_{RO2+RO2}[RO_2][RO_2]^T + k_{uni}[RO_2] \qquad (Eq.4)$$

Combing Eq.2-4, one can get Eq. 5:

$$\frac{\frac{d[ROH]}{dt} + \frac{d[ROOH]}{dt}}{\frac{d[R=O]}{dt}} = \frac{\alpha k_{RO2+RO2}[RO_2][RO_2]^T + k_{RO2+HO2}[RO_2][HO_2]\beta}{(1-\alpha)k_{RO2+RO2}[RO_2][RO_2]^T + k_{uni}[RO_2]}$$

$$= \frac{\alpha k_{RO2+RO2}[RO_2]^T + k_{RO2+HO2}[HO_2]\beta}{(1-\alpha)k_{RO2+RO2}[RO_2]^T + k_{uni}} \qquad (Eq. 5)$$

where $[RO_2]^T$ and $[HO_2]$ are the total concentrations of RO$_2$ and the concentration of HO$_2$ in the reaction system, respectively. $k_{uni}$, $k_{RO2+RO2}$ and $k_{RO2+HO2}$ represent the rate coefficient of the unimolecular termination of RO$_2$ and the bimolecular reactions of RO$_2$ with RO$_2$ and HO$_2$. α and 1-α are the carbonyl yield and the alcohol yield in reactions of RO$_2$ + RO$_2$, respectively. Wall loss was neglected due to its minor effects on the concentrations of the products (~5.8%).

For $C_{10}H_{15}O_x\bullet$ family, the Eq. 5 is equivalent to Eq. 6:

$$\left(\frac{\frac{d[ROH]}{dt} + \frac{d[ROOH]}{dt}}{\frac{d[R=O]}{dt}}\right)_{C_{10}H_{15}O_x\bullet} = \frac{d[C_{10}H_{16}O_x]_{ROH+ROOH}}{d[C_{10}H_{14}O_x]} = \frac{d[C_{10}H_{16}O_x] \times (1-[R=O]\%)}{d[C_{10}H_{14}O_x]} \qquad (Eq. 6)$$

For $C_{10}H_{17}O_x\bullet$ family, the Eq. 5 is equivalent to Eq. 7:

$$\left(\frac{\frac{d[ROH]}{dt} + \frac{d[ROOH]}{dt}}{\frac{d[R=O]}{dt}}\right)_{C_{10}H_{17}O_x\bullet} = \frac{d[C_{10}H_{18}O_x]}{d[C_{10}H_{16}O_x]_{R=O}} = \frac{d[C_{10}H_{18}O_x]}{d[C_{10}H_{16}O_x] \times [R=O]\%} \qquad (Eq. 7)$$

As shown in the Eq. 2, we assume the same $k_{uni}$, $k_{RO2+RO2}$, $k_{RO2+HO2}$, $\alpha$ and $\beta$ for a given $C_{10}H_{15}O_x\bullet$ and $C_{10}H_{17}O_x\bullet$ family. Consequently, the ratios of $\frac{d[ROH]+d[ROOH]}{d[R=O]}$ for $C_{10}H_{15}O_x\bullet$ and $C_{10}H_{17}O_x\bullet$ are the same. Eq. 8 can then be derived from Eq. 6 and 7, to yield [R=O]%, which represents the contribution of the carbonyls produced from $C_{10}H_{17}O_x\bullet$ to $C_{10}H_{16}O_x$.

$$\frac{d[C_{10}H_{16}O_x]\times(1-[R=O]\%)}{d[C_{10}H_{14}O_x]} = \frac{d[C_{10}H_{18}O_x]}{d[C_{10}H_{16}O_x]\times[R=O]\%} \tag{Eq. 8}$$

Based on this, about 90.1% and 98.8% of $C_{10}H_{16}O_x$ were estimated to be carbonyls from $C_{10}H_{17}O_x\bullet$ at low and high NO, respectively. As HOM in this study is likely formed via a 6-membered carbon ring opening as discussed below, most HOM $RO_2$ are likely primary or secondary $RO_2$ as shown in Scheme 1 and Scheme 2 and $RO_2$ distribution in Fig. S9. For primary and secondary HOM $RO_2$, although carbonyl yield and alcohol yield does not necessarily equal to 1, they are most likely to be 1 according to Jenkin et al (2019). With these equations, we can estimate carbonyl fractions formed via $C_{10}H_{17}O_x\bullet$ under

reasonable assumption. We did a sensitivity analysis to test the influence of varying the $k_{uni}$, $k_{RO2+RO2}$, $k_{RO2+HO2}$, and $\alpha$ using the ranges of these parameters reported in the literature on the fraction of carbonyl in $C_{10}H_{16}O_x$ and on the importance of H-abstraction channel in HOM formation. When $k_{uni}$, $k_{RO2+RO2}$, $k_{RO2+HO2}$, and $\alpha$ were varied in the range of $(0.01-1)*10^{-12}$ cm$^3$ molecule$^{-1}$ s$^{-1}$, $(0.001-1)*10^{-10}$ cm$^3$ molecule$^{-1}$ s$^{-1}$, $(0.5-2)*10^{-11}$ cm$^3$ molecule$^{-1}$ s$^{-1}$ based on the values in the literature (Crounse et al., 2013; Berndt et al., 2018; Ziemann and Atkinson, 2012), and 0.5, respectively, one can get the yield of carbonyl

according to Eq. 6 and Eq. 7, which ranged from 90%-96% at low NO and 97%-100% at high NO. This indicated that the yields of carbonyl are not sensitive to these assumptions of k and $\alpha$.

## 2.4 Theoretical kinetic study of $C_{10}H_{15}O$ alkoxy and $C_{10}H_{15}O_2$ peroxy radicals

The formation mechanism of $C_{10}H_{15}O$ alkoxy and $C_{10}H_{15}O_2$ peroxy radicals were considered in the theoretical kinetic study. The geometries of the intermediates and transition states for the first steps in the mechanism were first optimized using the

M06-2X/cc-pVDZ methodology (Zhao and Truhlar, 2008; Dunning, 1989), with an exhaustive characterization of all conformers for each reactant and transition state. All geometries obtained thus were further optimized at the M06-2X-D3/aug-cc-pVTZ level of theory which includes D3 diffusion corrections (Goerigk et al., 2017; Grimme et al., 2011). Moments of inertia for molecular rotation, and wavenumbers for vibration were obtained at the same level of theory, with a vibrational scaling factor of 0.971 (Alecu et al., 2010; Dunning, 1989). The barrier heights were further improved by single-point

calculations at the CCSD(T)/aug-cc-pVTZ level of theory (Dunning et al., 2001; Bartlett and Purvis, 1978) (all T1 diagnostics $\leq 0.029$). The expected uncertainty on the reaction barrier heights at this level of theory is $\pm 0.5$ kcal mol$^{-1}$. All quantum chemical calculations were performed using the Gaussian-16 software suite (Frisch et al., 2016). The quantum chemical data underlying the theoretical kinetic calculations is provided in the supplement.

The rate coefficients for the individual reactions at the high-pressure limit were then calculated using multi-conformer

transition state theory, MC-TST (Vereecken and Peeters, 2003), incorporating the data for all conformers obtained as described above. Tunnelling is accounted for using an asymmetric Eckart barrier correction (Johnston and Heicklen, 1962; Eckart, 1930). Based on earlier work at a similar level of theory in comparison with experimental data on H-migration in $RO_2\bullet$ radicals with no or only one oxygenated functionality (Vereecken and Nozière, 2020; Nozière and Vereecken, 2019) and available theoretical literature data on ring closure reactions (Vereecken et al., 2021), we estimate the thermal rates to be accurate to a

factor 2 to 3.

## 2.5 SAR-based mechanism development for autoxidation

The kinetics of the chemistry following the ring breaking is expected to be fairly well described based on structure-activity relationships (SARs), and no explicit theoretical kinetic calculations were performed. We employ the same approach for deducing the mechanism as described in our recent work (Shen et al., 2022). Briefly, we only take into account a limited

oxidation network by considering all possible reaction channels and select only the dominant channels, based on their rate as

predicted by SARs. For the rate coefficients for most H-migrations in RO$_2$• radicals, we base ourselves on the SAR by Vereecken and Nozière (2020); this SAR was reported to reproduce the scarce experimental data within a factor of 2, but for multi-functionalized species such as studied in this work the scatter on the data within each SAR category reaches an order of magnitude. For H-migrations in cycloperoxides, we additionally rely on the systematic study by Vereecken et al. (2021), who explicitly calculated rate coefficients for peroxy radicals formed after RO$_2$• ring closure reactions. For those reaction classes that are not covered by either SAR, we estimate a rate by extrapolating the reactivity trends in the SARs, albeit with a large uncertainty. For ring closure reactions in unsaturated RO$_2$• we employ on the SAR by Vereecken et al. (2021), where it is assumed that the presence of another cycloperoxide ring does not influence the rate. In assessing the fate of an RO$_2$• radical with one or more -OOH groups, we account for the possibility of H-atom scrambling, as described extensively in the literature (Praske et al., 2019; Møller et al., 2019; Jørgensen et al., 2016; Nozière and Vereecken, 2019; Knap and Jørgensen, 2017). These fast H-migrations between the -OO• and -OOH groups are typically much faster than other unimolecular channels, leading to a fast equilibration among all accessible OOH-substituted RO$_2$• radicals, and the dominant reaction is chosen among those available to the pool of RO$_2$• radicals. We refer to Vereecken and Nozière (2020) for a more detailed description of this feature. In the early stages of the oxidation, we also use the recent work by Piletic and Kleindienst (2022). Finally, for alkoxy radical chemistry, we employ the SARs for decomposition and H-migration by Novelli et al. (2021) and Vereecken and Peeters (2009, 2010). For hydroperoxyl-substituted alkoxy radicals we note explicitly that H-migration of the hydroperoxide H-atom, forming an alcohol and an RO$_2$ radical, is typically very fast, k(298 K) $\geq 10^{10}$ s$^{-1}$ (Vereecken and Nozière, 2020), and is often the most likely loss process in later alkoxy stages in the autoxidation chain. This fast H-migration supports competitive autoxidation even under high-NO conditions despite the formation of alkoxy intermediates that threaten to fragment the molecule.

## 3 Results and discussion

### 3.1 Overview of HOM spectra

The mass spectra of gas phase products HOM formed in the oxidation of limonene by OH at two different NO$_x$ levels are demonstrated in the Fig. 1. The HOM products can be classified according their mass to charge ratio (m/z) as either monomers (200-400 Th, including C$_6$-C$_9$ monomers and C$_{10}$ monomers (320-400 Th)) or dimers (480-600 Th, C$_{17}$-C$_{20}$ dimers). We did not observe any trimer products (Fig. S4). The signal intensity of monomers is higher than dimers at both low and high NO (Fig. 1 and Fig. 2), where monomers accounted for over 80% of total HOM. Both the signal intensity and the fraction of the observed dimers at high NO were much less than that at low NO (Fig. 1 and Fig. 2). In the following (Sect. 3.2) we discussed product distribution and formation mechanism of monomers and dimers in details.

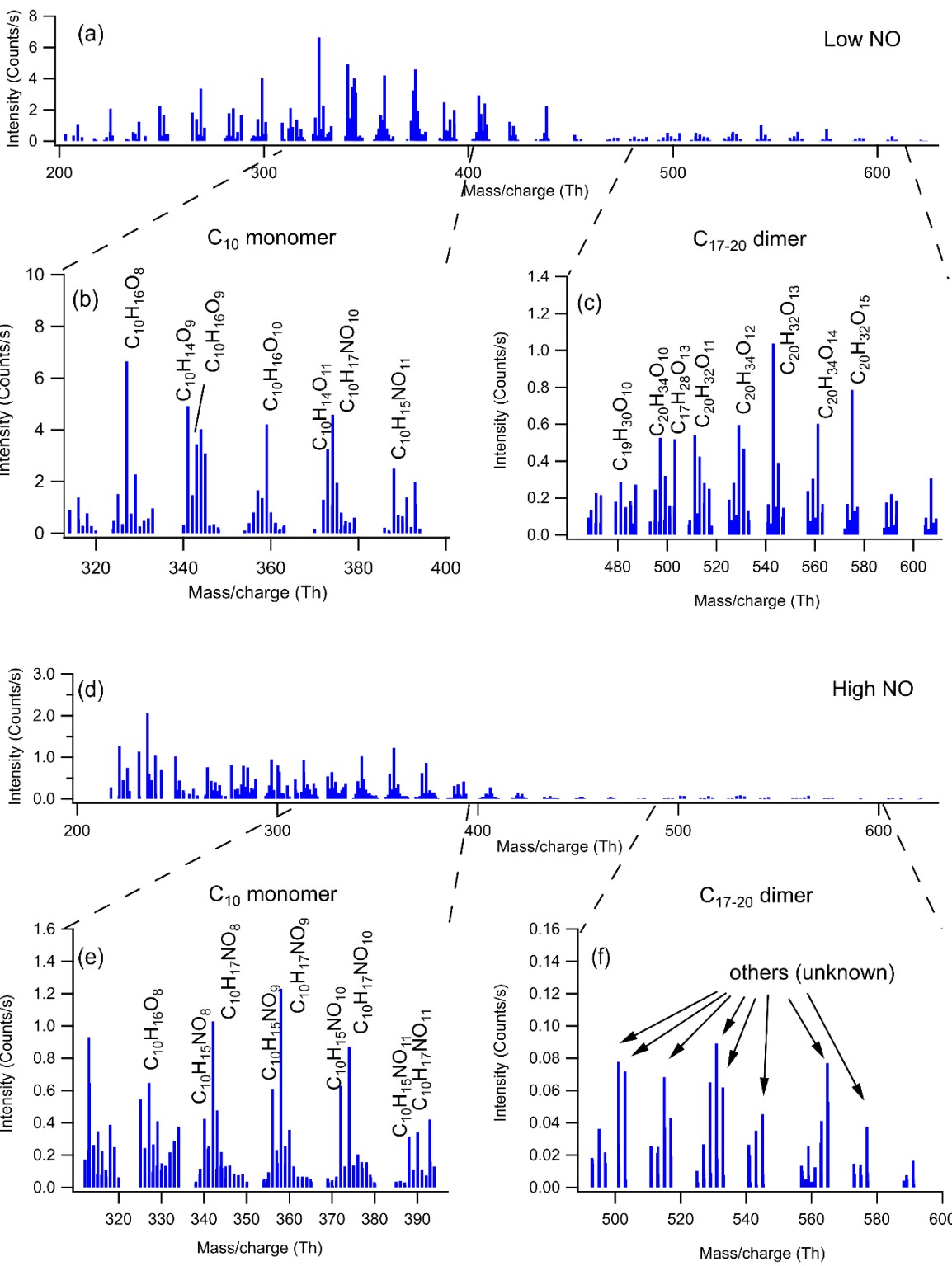

**Figure 1.** Mass spectrum of the HOM formed in the oxidation of limonene by OH in (a) low and (d) high NO condition. The mass spectra were obtained within 15 min after opening the louvres and were averaged over 15 min. Figure 1 (b, c, e, f) present expanded mass spectra where major peaks are labelled with their molecular formulas.

275

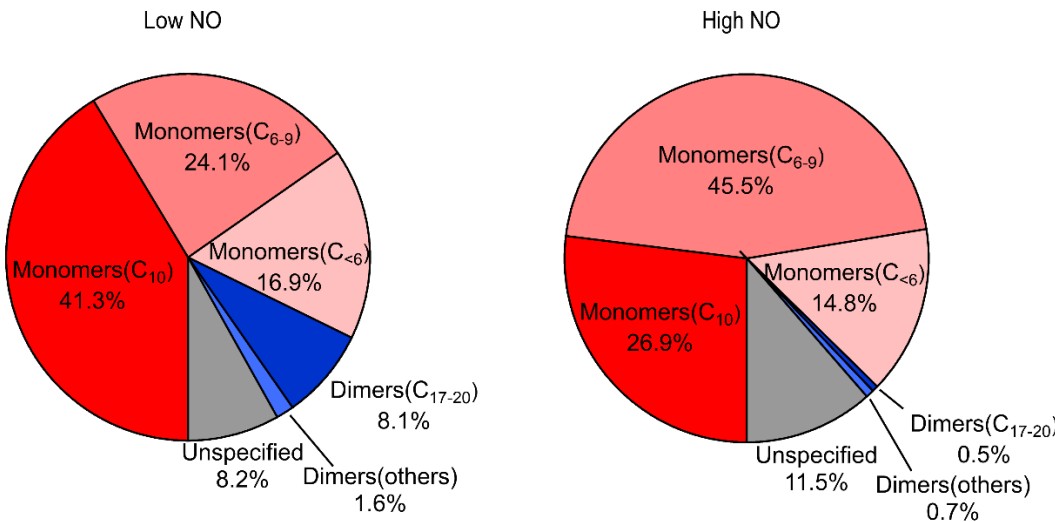

**Figure 2.** HOM product distribution including monomers ($C_{10}$, $C_{6-9}$, $C_{<6}$), dimers ($C_{17}-C_{20}$, other), and unspecified species in low and high NO conditions at the first 15 min of experiments.

### 3.2 Identification and formation mechanism of HOM

#### 3.2.1 Overview of HOM monomers

Closed-shell HOM monomers are characterized by repeating pattern of molecular formulas with an increasing number of oxygen atoms in the mass spectrum. The repeating pattern in the mass spectrum at every 16 Th form series of HOM monomer families, such as $C_6H_{8/10}O_x$, $C_6H_{9/11}NO_x$, $C_7H_{8/10/12}O_x$, $C_7H_{9/11}NO_x$, $C_8H_{10/12/14}O_x$, $C_8H_{11/13}NO_x$, $C_9H_{12/14/16}O_x$, $C_9H_{13/15}NO_x$, $C_{10}H_{14/16/18}O_x$, and $C_{10}H_{15/17}NO_x$ (Fig. S5). It is worth noting that some open-shell radicals, such as $C_{10}H_{15}O_x\bullet$(x=6-15) and $C_{10}H_{17}O_x\bullet$(x=6-15), were also observed; these are a significant fraction of HOM monomers and will be discussed in detail in section 3.2.2. These open-shell molecules were likely $RO_2$ radicals, as we are not able to detect the very short lived alkoxy radicals (RO) and alkyl radicals (R). The closed shell products ($C_{\leq10}H_yO_x$ and $C_{\leq10}H_yNO_x$) will be discussed in the following from the perspective of $RO_2$ chemistry, i.e., as termination products of $RO_2$. It should be noted that each molecular mass can represent several isomers, and that there may be more than one pathway to form a given product; our experiments do not allow to discriminate these and an in-depth discussion is outside the scope of this work.

HOM monomers were classified into $C_{10}$ and $C_{6-9}$ monomers according to the numbers of carbon atoms. The abundance of $C_{6-9}$ monomers (~27.0 % combined) was lower than $C_{10}$ compounds (~46.4 %) at low NO. In addition to $C_{10}$ and $C_{6-9}$ monomers, other monomers accounted for ~18.9% of all monomers and unspecified monomers accounted for ~7.7%. As shown in Fig. S5, $C_{6-10}$ monomers contain either none or one nitrogen atoms, and a higher fraction of organic nitrates was observed in high NO condition (nitrate HOM: 31% at low NO and 41% at high NO). This is as expected given that the fraction of $RO_2$ terminated by NO forming organic nitrate was higher at high NO. Higher fractions of organic nitrate have also been observed by the previous studies in the photo-oxidation of other monoterpenes at high NO (Pullinen et al., 2020; Shen et al., 2022).

#### 3.2.2 $C_{10}$ monomer product distribution

At low and high NO condition, $C_{10}$ compounds ($C_{10}H_{14}O_x$, $C_{10}H_{15}NO_x$, $C_{10}H_{16}O_x$, $C_{10}H_{17}NO_x$ and $C_{10}H_{18}O_x$) were the abundant monomers with a fractional contribution of ~27-~41% (see Fig. 3). In this study, we observed two major $RO_2$ radical families, $C_{10}H_{15}O_x\bullet$(x=6-15) and $C_{10}H_{17}O_x\bullet$(x=6-15), and their corresponding termination products, $C_{10}H_{14}O_{x-1}$, $C_{10}H_{16}O_{x-1}$, $C_{10}H_{16}O_x$, and $C_{10}H_{15}NO_{x+1}$, and $C_{10}H_{16}O_{x-1}$, $C_{10}H_{18}O_{x-1}$, $C_{10}H_{18}O_x$, and $C_{10}H_{17}NO_{x+1}$ respectively, which contained carbonyl, hydroxyl, hydroperoxyl and nitrate, respectively (see Fig. 3, Fig. S5, Table S2, Table S3, Table S6, and Table S7). $C_{10}H_{14}O_x$ and $C_{10}H_{16}O_x$

were likely carbonyl compounds formed via unimolecular termination of $C_{10}H_{15}O_x\bullet$ and $C_{10}H_{17}O_x\bullet$, respectively. Overall, at high NO, organic nitrates were the most abundant among all classes of HOM monomers and their relative proportion was much higher than at low NO (Fig. 3). Under low NO condition, $C_{10}H_{14}O_x$ and $C_{10}H_{16}O_x$ family are the two most abundant family among $C_{10}$ monomers. This is similar to the $C_{10}$ monomers products from photooxidation of α-pinene by OH radical (Shen et al., 2022).

310

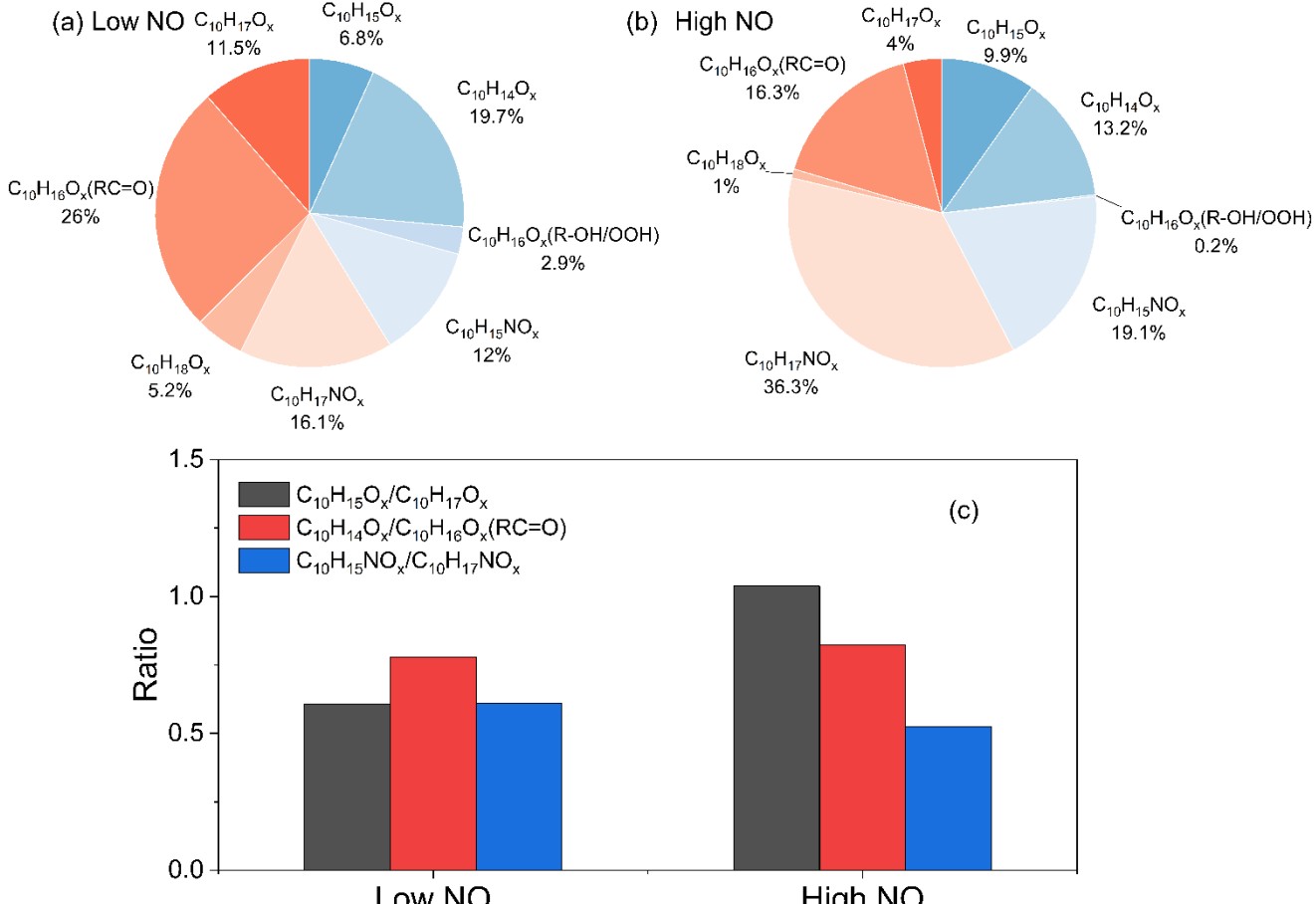

**Figure 3.** Product distribution within $C_{10}$ monomers at (a) low and (b) high NO and (c) the ratio of $C_{10}H_{15}O_x\bullet$ to $C_{10}H_{17}O_x\bullet$ and of their termination products. As in Fig.2, data of the first 15 min of the experiments were used.

### 3.2.3 Monomers with $C_{<10}$ monomers and dimers

A number of monomers with carbon number less than 10 ($C_{<10}$) were observed in this study. The monomers ($C_{<10}$) account for about 24.1-45.5% of the total monomers. The two radicals $C_7H_9O_x\bullet$ and $C_7H_{11}O_x\bullet$ were likely produced from $C_{10}H_{15}O_x\bullet$ radicals and $C_{10}H_{17}O_x\bullet$ radicals, respectively (see below). The ratios of total termination products from $C_7H_9O_x\bullet$ to those from $C_7H_{11}O_x\bullet$ are 0.7 at both low and high NO, which is generally consistent with the ratio of $C_{10}H_{15}O_x\bullet$ radicals and $C_{10}H_{17}O_x\bullet$ radicals for $C_{10}$-related products. This suggests that chemistry related to $C_{10}H_{15}O_x\bullet$ radicals plays a significant role in formation of $C_{<10}$ monomers. Although the concentrations of monomers ($C_{<10}$) are smaller than of $C_{10}$ monomers, they may also contribute to the formation of dimers, which further condense onto particles.

We observed a number of dimer families, including $C_{20}H_{30}O_x$ (x=10,12-18), $C_{20}H_{32}O_x$ (x=9-18), $C_{20}H_{34}O_x$ (x=10-17), $C_{19}H_{32}O_x$ (x=10-15), $C_{19}H_{31}NO_x$ (x=11-18), $C_{19}H_{30}O_x$ (x=10-17), $C_{18}H_{30}O_x$ (x=10-14), $C_{19}H_{28}O_x$ (x=11-15), $C_{17}H_{26}O_x$ (x=11-15), $C_{17}H_{24}O_x$ (x=12-17) in low NO condition, and $C_{20}H_{30}O_x$ (x=12-16), $C_{20}H_{32}O_x$ (x=11-18), $C_{20}H_{34}O_x$ (x=10-15), $C_{19}H_{28}O_x$ (x=11-17), $C_{17}H_{26}O_x$ (x=12-15) in high NO condition (see Table S4 and Table S5). Compared to monomers, the peak intensity of dimers both at low and high NO conditions are much lower (Fig. S7 and Fig. S8). Furthermore, the peak intensity of dimers

in low NO condition is higher than that in high NO condition. This inhibitory effect of $NO_x$ in dimer formation is consistent with the phenomenon described in previous studies (Pullinen et al., 2020; Yan et al., 2016; Shen et al., 2022). In the following discussion we will focus on $C_{10}$ monomers. However, more information on $C_{<10}$ monomers and dimers can be found in Supplement Sect. S2.

### 3.3 Formation pathways of HOM monomers

In this section we discuss potential formation pathways for $C_{10}$ monomers. These pathways are tentative only, and are based on the current knowledge on autoxidation as represented in various SARs, where we examine only those paths estimated as being the fastest. Our observations do not distinguish between different isomers and we cannot validate the reaction scheme. We also observed HOM with only 7, 8 or 9 carbon atoms and their possible formation pathways, involving a fragmentation step in an intermediate alkoxy radical, are inspected in the supporting information (see schemes S1 and S2).

### 3.3.1 HOM from OH addition

The $C_{10}H_{17}O_x\bullet$ radicals are formed from the OH addition reaction of limonene and subsequent autoxidation (see Table S3). Scheme 1 shows possible oxidation routes for the OH addition to both double bonds, considering the particular channels leading to tertiary alkyl radicals and depicting the most likely reactions as estimated based on theoretical work and SARs (Peeters et al., 1999; Vereecken et al., 2007; Jenkin et al., 2018). The $C_{10}H_{17}O_3\bullet$ $RO_2$ radical formed after $O_2$ addition on the OH adducts has access to a wide range of reactions, but in most cases the six-membered ring is retained in the first $RO_2$ steps, leading to slow autoxidation steps (Vereecken et al. 2021). This allows the reaction with NO to become important, whereafter the decomposition of a β-OH or β-OOH alkoxy radical leads to carbonyl formation and thus chain termination. This analysis suggests that HOM formation after OH addition either proceeds through less favourable pathways or involves secondary chemistry. The complete explicit mechanism is highly branched and complex, and missing pathways in Scheme 1 could together contribute to sizable HOM formation. Furthermore, the rate estimates are tentative, with uncertainties of an order of magnitude or more, so we cannot exclude that primary HOM formation is possible and important. The primary products predicted in Scheme 1 (or formed through similar termination reactions in other channels) can react again with OH, initiating secondary autoxidation. Most of these products have the six-membered ring broken, and should be more amenable to further autoxidation steps and thus readily yield HOM. However, this extra bimolecular OH reaction takes time, and can delay formation of HOM. An analysis of the secondary chemistry is outside the scope of this work, and at this time we do not propose a reaction scheme that covers the full range of $C_{10}H_{17}O_x\bullet$ radicals and related products observed.

In our previous study, the average bimolecular $RO_2$ loss rate was estimated at ~0.02 $s^{-1}$ (low NO) and ~3.5 $s^{-1}$ (high NO) (Zhao et al., 2018a). In both cases $RO_2$ loss is due predominantly to the reaction with NO (see Fig. S3), which leads to the organic nitrate termination products, as illustrated in Fig. 3. Even at low NO with ~0.2 ppb, organic nitrates contribute a large portion of HOM (33%). The relevant termination products from $C_{10}H_{17}O_x\bullet$ (x=8-13) in reactions with $HO_2$ and $RO_2$ are identified as $C_{10}H_{18}O_x$, $C_{10}H_{18}O_{x-1}$, and $C_{10}H_{16}O_{x-1}$. However, $C_{10}H_{18}O_x$ accounted for only 5.2% and 1% in $C_{10}$ monomers at low and high NO, respectively, as in high NO conditions these loss processes are overwhelmed by the $RO_2$ + NO reaction.

**Scheme 1.** Example reaction scheme for $C_{10}H_{17}O_x\bullet$ RO$_2$ formation after OH-addition in limonene, based on the most likely reactions predicted in theoretical work and SAR predictions. All RO$_2$ intermediates have competing reactions (not shown) under current conditions with HO$_2$ (forming hydroperoxides) and NO (forming alkoxy and nitrates).

### 3.3.2 HOM from H-abstraction by OH

The observed $C_{10}H_{15}O_x\bullet$ radicals mainly originate from an H-abstraction reaction by OH, removing one of the limonene H-atoms. In principle, $C_{10}H_{15}O_x\bullet$ peroxy radicals might also form through secondary chemistry of first-generation $C_{10}$ oxidation products of the limonene+OH reaction. The limonaldehyde ($C_{10}H_{16}O_2$) is the most abundant (99%) first-generation $C_{10}$ product reported in limonene+OH reaction (Hakola et al., 1994; Larsen et al., 2001), which can form $C_{10}H_{15}O_4\bullet$ and the $C_{10}H_{15}O_x\bullet$ family by further autoxidation through H-abstraction and subsequent O$_2$ addition. Therefore, we take limonaldehyde into account as the most competitive candidate. For the early stages of our experiments (first 15 min), however, we find that secondary chemistry is not important (Section S2 and Fig. S9 in Supplement). An example pathway for $C_{10}H_{15}O_x\bullet$ RO$_2$ radical formation is shown in Scheme 2, based on the most facile H-abstraction channel from limonene. Direct autoxidation of the

nascent $RO_2$ is slow, k=~$10^{-2}$ s$^{-1}$, and formation of an alkoxy radical is to be expected immediately or after very few autoxidation steps, especially in high NO conditions. Once the ring structure is broken, fast autoxidation steps are accessible. All $RO_2$ intermediates have competing reactions (not shown) under current conditions with $HO_2$ (forming hydroperoxides) and NO (forming alkoxy radicals and nitrates). Alkoxy radicals formed thus can fragment, or continue autoxidation after ring

380  breaking or fast migration of an hydroperoxide H-atom, forming a wider variety of HOM. The nascent peroxy radical formed after H-abstraction of the allylic tertiary H-atom in limonene is not amenable to efficient autoxidation and HOM formation, due to the geometric constraints in the ring. Theoretical kinetic calculations (see Table 1) show that the fastest autoxidation steps occur at rates of ~$2\times10^{-2}$ s$^{-1}$, i.e., of similar magnitude as the $RO_2$ loss with NO in the low-NO experiments, but negligible against the $RO_2$ + NO rate at high NO. Our rate predictions agree with the recent theoretical study by Piletic and Kleindienst

385  (2022). Further autoxidation steps with the ring structure intact are expected to be unfavourable, as well as autoxidation initiated through other H-abstraction sites (Vereecken et al., 2021; Piletic and Kleindienst, 2022). The reaction of the primary $RO_2$ with NO, however, leads to an alkoxy radical that is most likely to break the six-membered ring, with a total rate coefficient exceeding ~$10^6$ s$^{-1}$ (see Table 1). This alkoxy-peroxy autoxidation step thus removes the geometric constraint, and fast autoxidation reactions become accessible, with rates enhanced by the presence of the double bonds (Vereecken and

390  Nozière, 2020; Møller et al., 2019). Though further alkoxy-peroxy steps are not necessary for HOM formation from limonene, subsequent autoxidation steps will compete against bimolecular reactions with NO, suggesting that more alkoxy-peroxy steps may occur even when the unimolecular lifetime of these later non-cyclic $RO_2$ radicals is typically shorter than that of the early-stage cyclic $RO_2$. Some of these alkoxy intermediates will preferentially break the carbon backbone, leading to fragmentation, while other alkoxy radicals will proceed by H-migration and continue the autoxidation chain. In the latter case, autoxidation

395  then no longer follows the systematic series of adding 2 O-atoms per autoxidation step, and the $C_{10}H_{15}O_x\bullet$ formation mechanism can proceed both by even-numbered and odd-numbered oxygen numbers x. At high NO concentrations, formation of alkoxy radicals becomes more likely, and thus fragmentation can become more prominent, reducing the yield of $C_{10}H_{15}O_x\bullet$ related HOM. This is in agreement with the current observations, where more fragment $C_{<10}$ monomers (47.2% in total monomers) are observed at high NO, than at low NO (27.0 %). Further autoxidation of the fragments is also possible, leading

400  to $C_{<10}$ monomers and dimer formation, as discussed in the supplement. The need of a ring-breaking alkoxy-peroxy step in the proposed HOM formation mechanism does suggest that the highest yields of limonene HOM formation may occur at slightly higher NO concentrations than for non-cyclic VOCs who autoxidize even without alkoxy steps.

**Table 1.** H-migration, ring closure and C-C bond scission reactions in $RO_2^{\bullet}$ and $RO^{\bullet}$ radicals formed from limonene H-abstraction, listing barrier heights $E_b$ (kcal mol$^{-1}$), rate coefficient k at 298 K (s$^{-1}$), and parameters for a Kooij equation fit $k(T)=A \times T^n \times \exp(-E_a/T)$ for the temperature range 200-450 K, calculated using the CCSD(T)//M06-2X-D3 level of theory.

| $RO_2^{\bullet}$ | Reaction type | $E_b$ | $k(298K)$ | $A$ / s$^{-1}$ | $n$ | $E_a$ / K |
|---|---|---|---|---|---|---|
| | 5-Ring closure (a) | 19.0 | $2.1 \times 10^{-2}$ | 5.40E+07 | 1.40 | 8840 |
| | 6-Ring closure (b) | 20.2 | $1.8 \times 10^{-3}$ | 6.35E+07 | 1.27 | 9390 |
| | Allyl-1,5-H-shift (c) | 24.4 | $1.2 \times 10^{-3}$ | 6.45E-91 | 32.94 | -3977 |
| | 4-Ring closure (d) | 26.9 | $2.4 \times 10^{-8}$ | 2.50E+03 | 2.79 | 12292 |
| | 5-Ring closure (e) | 29.8 | $2.3 \times 10^{-10}$ | 1.04E+03 | 2.83 | 13489 |
| | Allyl-1,5-H-shift (f) | 22.5 | $2.0 \times 10^{-2}$ | 8.68E-81 | 29.51 | -3667 |
| | Other H-migrations lead to high ring strain and are not competitive | | | | | |
| | Ring opening (a) | 9.4 | $9.5 \times 10^5$ | 7.60E+08 | 1.42 | 4409 |
| | Ring opening (b) | 10.5 | $1.6 \times 10^5$ | 1.01E+09 | 1.39 | 4972 |
| | Fragmentation (c) | 17.4 | $3.4 \times 10^0$ | 2.49E+09 | 1.48 | 8602 |
| | H-migrations or ring closure leads to high ring strain and are not competitive | | | | | |

**Scheme 2.** Example reaction scheme for HOM formation after H-abstraction in limonene, based mostly on SAR prediction and starting at the fastest of 5 allylic H-abstraction sites.

### 3.3.3 Relative contribution of OH addition versus OH H-abstraction

As $C_{10}H_{15}O_x\bullet$ (x=6-15) is formed via the OH H-abstraction pathway and $C_{10}H_{17}O_x\bullet$ is formed via OH addition, we can compare the relative importance of these two pathways via the ratios of $C_{10}H_{15}O_x\bullet$ to $C_{10}H_{17}O_x\bullet$ radicals and their termination products. At low NO, the ratio of the concentrations of $C_{10}H_{15}O_x\bullet$ to $C_{10}H_{17}O_x\bullet$ was 0.61 and the ratios of the concentrations of $C_{10}H_{15}O_x\bullet$ related termination products carbonyls ($C_{10}H_{14}O_x$) and organic nitrates ($C_{10}H_{15}NO_x$) to $C_{10}H_{17}O_x\bullet$ related termination products carbonyls ($C_{10}H_{16}O_x$) and organic nitrates ($C_{10}H_{17}NO_x$) are 0.78 and 0.61, respectively (Fig. 3c). At high NO, the ratio of the concentrations of $C_{10}H_{15}O_x\bullet$ to $C_{10}H_{17}O_x\bullet$ was ~1.1 (Fig. 3) and the corresponding ratios of the concentrations of $C_{10}H_{15}O_x\bullet$ related termination products to $C_{10}H_{17}O_x\bullet$ related termination products are 0.82 and 0.53, respectively (Fig. 3c). Note that $C_{10}H_{16}O_x$ can be termination products from either $C_{10}H_{15}O_x\bullet$ or $C_{10}H_{17}O_x\bullet$ radicals depending on whether they contain hydroxyl/hydroperoxyl (ROH/ROOH) or carbonyl (RC=O) functionalities; the relative substituent contributions in $C_{10}H_{16}O_x$ were quantified using the method described in Sect. 2.2. The concentrations of $C_{10}H_{16}O_x$ and $C_{10}H_{18}O_x$ (alcohol and hydroperoxide products), formed from $C_{10}H_{15}O_x\bullet$ and $C_{10}H_{17}O_x\bullet$ radicals respectively, were found to be negligible, and we can assign $C_{10}H_{16}O_x$ products as carbonyls formed from the OH addition $C_{10}H_{17}O_x\bullet$ radicals. The $NO_x$ dependence for $C_{10}H_{15}O_x\bullet/C_{10}H_{17}O_x\bullet$ may be attributed to the differences in their reactivity. One explanation for the $NO_x$ dependence is that the autooxidation of $C_{10}H_{15}O_x\bullet$ $RO_2$ radicals may be faster than that of $C_{10}H_{17}O_x\bullet$ $RO_2$ radicals, which leads to the lower concentration of $C_{10}H_{15}O_x\bullet$ at high NO and thus higher sensitivity to NO concentrations. Ratios of $C_{10}H_{16}O_x$(R-OH/OOH)/$C_{10}H_{15}O_x\bullet$ can be derived of 0.47 and 0.02 at low and high NO, respectively. The decrease of $C_{10}H_{16}O_x$(R-OH/OOH)/$C_{10}H_{15}O_x\bullet$ at high NO compared to low NO was more evident than the decrease of $C_{10}H_{18}O_x/C_{10}H_{17}O_x\bullet$. Theoretically, though, they should be similar. The difference may be attributed to the shift in $C_{10}H_{15}O_x\bullet$ distribution with different number of O, as evident in Fig. S6, and different isomers at high NO compared to low NO. At high NO there might thus be more $C_{10}H_{15}O_x\bullet$ that react slower with $HO_2$ or have a lower branching ratio forming ROOH in $RO_2+HO_2$, which depends on the explicit $RO_2$ structure, or have a lower yield forming ROH in $RO_2+RO_2$. Overall, the $C_{10}H_{15}O_x\bullet$ related products formed via the H-abstraction channel contribute 41% and 42% of $C_{10}$ HOM monomer respectively at low and high NO. The ratio between products of $C_{10}H_{15}O_x\bullet$ and $C_{10}H_{17}O_x\bullet$ radicals is stable at both low and high NO (Fig. 4), except for a small decrease at the early reaction times under low NO conditions (Fig. 4c), albeit with large error. At the same time, the nitrate ratio $C_{10}H_{15}NO_x/C_{10}H_{17}NO_x$ is lower than that of $C_{10}H_{14}O_x/C_{10}H_{16}O_x$, which indicates that carbonyl production from $C_{10}H_{15}O_x\bullet$ is more efficient than from $C_{10}H_{17}O_x\bullet$. The concentration of peroxy radicals $C_{10}H_{15}O_x\bullet$ and the related termination products $C_{10}H_{14}O_x$ and $C_{10}H_{15}NO_x$ were comparable to $C_{10}H_{17}O_x\bullet$ and its corresponding products, illustrating the significant role of the H-abstraction pathway in the HOM formation from limonene oxidation by OH at low and high NO. The much higher abundance of carbonyl than alcohol is unlikely to be explained by the $RO+O_2$ forming carbonyl as for large RO ($C_{10}$ in this study), the $RO+O_2$ is generally slower than unimolecular reactions including the isomerization (H-shift, i.e., alkoxy-peroxy pathway) and decomposition. The higher abundance of carbonyl products compared to alcohol products indicates that here a large fraction of the carbonyls are not formed from $RO_2 + RO_2$ reactions (see also Fig. S3), but rather from termination reactions in HOOQOO• radicals eliminating an OH radical after an α-OOH H-atom migration, forming O=QOOH. This observation is in agreement with recent findings for α-pinene (Shen et al., 2022) and previous studies (Miller et al., 2005; Taatjes, 2006; Rissanen et al., 2014; Bianchi et al., 2019).

The large contribution of $C_{10}H_{15}O_x\bullet$ related products to HOM formation can be attributed to the significant contribution of H-abstraction by OH in the initial step. While OH addition is believed to be the major initiation channel in the reaction of limonene+OH and H-abstraction is often ignored in current chemical mechanisms (e.g., MCM v3.3.1), H-abstraction by OH is clearly not negligible. Already previous studies have suggested that H-abstraction can be a significant reaction pathway with ~34±8% branching ratio for the reactions of limonene with OH radical at 298 K (Rio et al., 2010; Dash and Rajakumar, 2015), and 33.6±4.8% at 293 K (Braure et al., 2014). We found that the ratio of $C_{10}H_{15}O_x\bullet$ to $C_{10}H_{17}O_x\bullet$ at high NO condition was higher than 2, emphasizing the importance of H-abstraction. A similar abundance of carbonyls ($C_{10}H_{14}O_x$) and organic nitrates

($C_{10}H_{15}NO_x$) stemming from $C_{10}H_{15}O_x\bullet$ radicals compared to their counterparts $C_{10}H_{16}O_x$ and $C_{10}H_{17}NO_x$ from $C_{10}H_{17}O_x\bullet$ at both low and high NO levels (Fig. 3) likewise indicates that H-abstraction in the first step of limonene oxidation by OH is important for the subsequent oxidation. Moreover, the high relative yields of HOM from the H-abstraction channel compared to its contribution in the limonene+OH initiation reaction can be tentatively attributed to the difference in the autoxidation mechanisms (Scheme 1 and 2). Many of the autoxidation channels following OH addition lead efficiently to termination

products (Scheme 1) and requires additional OH reactions to undergo further oxidation, whereas the $RO_2$ formed from H-abstraction readily lend themselves for a sequence of $O_2$ additions, once the ring structure is broken. The key difference is the presence of β-OH and β-OOH moieties in the alkoxy in the OH addition branch, which can form α-OH alkyl radicals and α-OOH alkyl radicals in fragmentation steps and further lead to $HO_2$ formation from α-OH alkyl radicals + $O_2$ (R8) or by regeneration of OH from α-OOH alkyl radicals (R9), thus preventing ring breaking without termination of the autoxidation

chain:

$$R_aR_bC^\bullet OH + O_2 \rightarrow R_aC(=O)R_b + HO_2 \qquad\qquad (R8)$$

$$R_aR_bC^\bullet OOH \rightarrow R_aC(=O)R_b + OH \qquad\qquad (R9)$$

    In contrast, the alkoxy step after H-abstraction breaks the 6-membered ring to a new $RO_2$, enhancing autoxidation (scheme 2). The propensity of the $RO_2$ from OH adducts to terminate the autoxidation chain, especially at higher NO where β-OH and

β-OOH alkoxy decomposition is prevalent, then allows the H-abstraction $RO_2$ to play a more dominant role. Contrary to our earlier studies on β-pinene or limonene with $NO_3$ (Shen et al., 2021; Guo et al., 2022), where we were able to assign $RO_2$ and products to first-, second-, or later-generation chemistry, no such clear n-th generation distinction can be made for the HOM mass spectrum traces in the limonene+OH system. This suggests that the observed HOMs are a mixture formed in several generations of OH-initiation reactions, and lends further credence to our proposal that termination reactions requiring

additional OH reactions are hampering the main OH addition channel from efficiently producing HOMs.

    To our knowledge, no previous studies have observed $C_{10}H_{15}O_x\bullet$ radicals in limonene oxidation by OH, although in ozonolysis the formation of $C_{10}H_{15}O_x\bullet$ and their termination products including $C_{10}H_{14-16}O_{6-10}$ monomers and $C_{18-20}H_{28-34}O_{6-16}$ was observed (Hammes et al., 2019; Tomaz et al., 2021), and field observations have shown the presence of $C_{10}H_{15}O_x\bullet$ radicals in the atmosphere (Yan et al., 2020; Massoli et al., 2018). Recent theoretical work from Piletic and Kleindienst (2022)

concluded that H-abstraction does not contribute to HOM in limonene+OH, but this study did not account for any alkoxy-peroxy autoxidation steps. Our study, then, shows for the first time that H-abstraction significantly contributes to HOM formation in the reaction of limonene+OH under reaction conditions that allow formation of alkoxy radicals such as by reaction with NO, $RO_2$ or $NO_3$, as commonly found in the atmosphere.

    Currently, an absolute calibration using HOM standards is not possible mainly due to the difficulty to synthesize pure HOM

and unclear chemical structures of many HOM. However, we think that it is reasonable to expect a generally similar sensitivity for HOM in this study for the following reasons. First, Hyttinen et al. (2017) found that the increase in binding energy with $NO_3^-$ for molecules with an additional hydroxyperoxy group to two hydrogen bond donor functional groups is small for HOM formed in cyclohexene ozonolysis. As HOM in this study generally contain more than two hydrogen bond donor functional groups, their sensitivity is expected to be similar. We used a unified $H_2SO_4$-based calibration coefficient for HOM, which is

commonly used to calibrate $NO_3^-$-CIMS (Kirkby et al., 2016; Jokinen et al., 2015; Rissanen et al., 2014; Ehn et al., 2014). Second, although underestimation of certain HOM $RO_2$ formed from α-pinene+OH reaction has been reported (Berndt et al., 2016), such underestimation was mainly attributed to the steric hindrance in forming HOM-nitrate cluster for HOM with bicyclic structures ($C_{10}H_{17}O_7\bullet$) ((Berndt et al., 2016), Section 3.2 of the Supplement therein) and thus such underestimation is not common for all HOM. In our study, we found the significance of $C_{10}H_{15}O_x\bullet$-related product at all oxygen

contents, particularly for closed-shell products with number of oxygen atom great than 8, indicative of more H-donating functional groups (Fig. S6 in revised Supplement). This indicates that the significance of $C_{10}H_{15}O_x\bullet$ related products is not affected by the detection sensitivity, which would mostly affect the sensitivity of less oxygenated compounds. And the

presence of NO particularly at high NO leads to ring-opening reactions as shown in Scheme 1. Therefore, the HOM products from OH addition in this study are likely to form stable clusters with nitrate and thus have similar sensitivity with HOM formed

via H-abstraction in nitrate CIMS. Third, our previous study showed that using an unified sensitivity of $H_2SO_4$ only leads to a maximum uncertainty of a factor of two by comparing the condensation HOM and corresponding increase of aerosol mass (Pullinen et al., 2020). If for some currently unknown reason $C_{10}H_{17}O_x\bullet$-related products had higher sensitivity than $C_{10}H_{15}O_x\bullet$-related products, this would lead to under-estimate of the significance of OH H-abstraction pathway. This will not change our conclusion that the $C_{10}H_{15}O_x\bullet$ related products contribute significantly to HOM formation.

We have further estimated the uncertainty of fraction of $C_{10}H_{15}O_x\bullet$-related products in C10-HOM resulted from the allocation of carbonyls and alcohols in $C_{10}H_{16}O_x$ in Eq.2-8. The contributions of $C_{10}H_{15}O_x\bullet$-related products range from 39.5% to 41.4% at low NO and 42.2% to 42.6% at high NO, respectively. We found that fraction of $C_{10}H_{15}O_x\bullet$ related products in $C_{10}$-HOM was not much affected by how carbonyls and alcohols in $C_{10}H_{16}O_x$ is allocated.

### 3.3.4 Comparison against α-pinene HOM formation

In our previous study, we have observed the large contribution of hydrogen abstraction OH to HOM formation in the oxidation of α-pinene by OH (Shen et al., 2022). There, at low NO (0.03-0.1 ppb), the ratio of HOM formed from hydrogen abstraction relative to OH addition increased in the beginning of the experiment and had a delay of 3-5 min before reaching 1:1. At high NO (~17 ppb), an enhancement of the HOM yields formed through H-abstraction was observed, both absolute and relative to

the HOM formed through OH addition. Alkoxy radical steps were found to be a prerequisite for the autoxidation and thus HOM formation in the hydrogen abstraction pathway. As discussed in Sect. 3.3.3, in limonene+OH oxidation, the ratios of $C_{10}H_{14}O_x$ to $C_{10}H_{16}O_x$ and of $C_{10}H_{15}NO_x$ to $C_{10}H_{17}NO_x$ and the time series of the ratios were similar at low and high NO. These results indicate that the rate of autoxidation for both the limonene H-abstraction and OH addition channels are affected to a similar extent by competing bimolecular reactions such as with NO. This contrasts with the enhancement of the HOM yield

through H-abstraction in the α-pinene system with increasing NO concentrations (Shen et al., 2022).

The different kinetic behaviour relative to competing reactions can be attributed to the different molecular structures of α-pinene and limonene. α-pinene is a bicyclic rigid molecule, and the $RO_2$ formed from H-abstraction do not undergo autoxidation as rate coefficients are $\leq 10^{-4}$ s$^{-1}$ (Shen et al., 2022). Even after breaking the first ring in an alkoxy step, autoxidation rates remain fairly slow, $\leq 10^{-1}$ s$^{-1}$, requiring a second alkoxy step with ring breaking before fast autoxidation

occurs. This makes HOM formation through H-abstraction sensitive to the NO concentration in the first few steps. In contrast, limonene has only a single 6-ring and only a single alkoxy ring breaking is needed to allow fast autoxidation in the resulting non-cyclic molecule. Moreover, the primary $RO_2$ from H-abstraction of limonene can still undergo some slow autoxidation, with rates ~$10^{-2}$ s$^{-1}$ (see Table 1), and possibly break the ring at a later stage (see Scheme 2). Overall, then, autoxidation in limonene system after H-abstraction is more competitive even at lower NO than α-pinene system. We should note, though,

that the low-NO experiments of limonene oxidation were performed at NO concentrations that were higher by a factor 2 to 7 compared to the low-NO experiment for α-pinene (~0.2 ppb and 0.03-0.1 ppb, respectively); we speculate that an NO-dependence of the ratio of HOM formed via H-abstraction versus OH addition might become apparent even in the limonene system at strongly reduced NO levels. The higher absolute contribution of HOM from the H-abstraction channel in limonene (H-abstraction related HOM yields: 0.77% low NO and 0.10% at high NO, shown in Sect. 3.4) is also affected by the higher

branching ratio of H-abstraction in limonene+OH, ~34%  (Rio et al., 2010; Dash and Rajakumar, 2015), compared to α-pinene+OH, ~11% (Vereecken and Peeters, 2000).

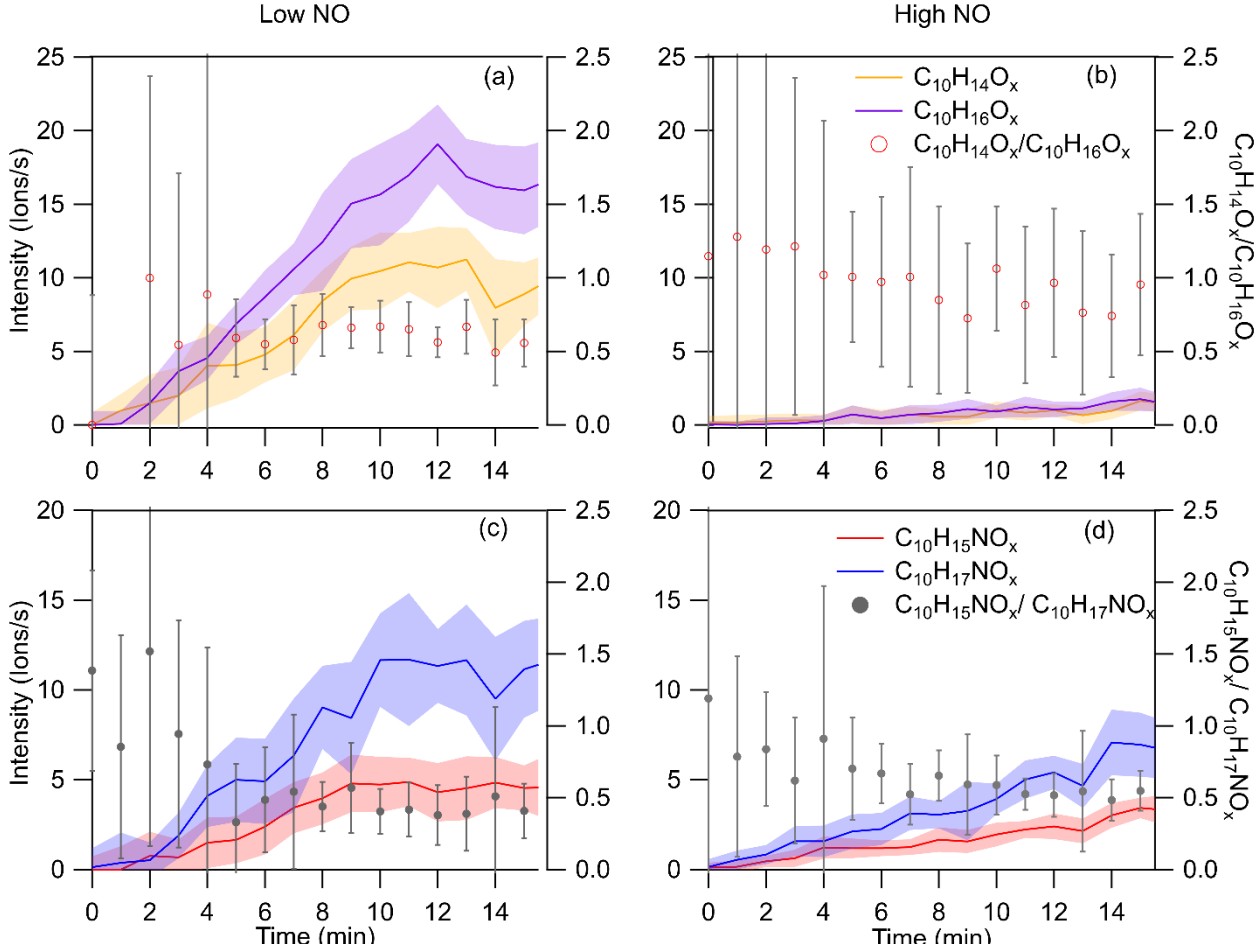

**Figure 4.** Time series of $C_{10}H_{15}O_x\bullet$ and $C_{10}H_{17}O_x\bullet$ related families of carbonyls and organic nitrates during the first 15 min of the experiments. The carbonyl families $C_{10}H_{14}O_x$ (yellow), $C_{10}H_{16}O_x$ (purple), and their concentration ratio ($C_{10}H_{14}O_x$ / $C_{10}H_{16}O_x$) are shown for **(a)** low and **(b)** high NO experiments. Time series for organic nitrate families $C_{10}H_{15}NO_x$ (red) and $C_{10}H_{17}NO_x$ (blue), and concentration ratio of $C_{10}H_{15}NO_x$ / $C_{10}H_{17}NO_x$, are similarly demonstrated in **(c)** for low NO and **(d)** high NO experiments. All data are averaged to 1 min and the error bar represents one standard deviation.

## 3.4 HOM yield

The HOM molar yields during the first 60 min was estimated to be $1.97^{+2.52}_{-1.06}\%$ and $0.29^{+0.38}_{-0.16}\%$ at low and high NO respectively. In brief, the uncertainty (-54%/+128%) was estimated from the HOM signal intensity, VOC concentration, and calibration coefficient of $H_2SO_4$ as discussed by Zhao et al (2021) (see Supplement S1). It is noted that the HOM yield is not very sensitive to the vapor wall loss rate, since HOM yields only change 11% to -6% upon an 100% increase and 50% decrease in wall loss rate (Zhao et al., 2021). We further note that these HOM yields may be subject to uncertainties due to the assumption that HOM have the same sensitivity as $H_2SO_4$ as we discussed in Sect. 3.3.3. As mentioned above, our previous study showed that using an unified sensitivity of $H_2SO_4$ only leads to a maximum uncertainty of a factor of two by comparing the condensation HOM and corresponding increase of aerosol mass (Pullinen et al., 2020). The lower HOM yields at high NO can be partly attributed to lower fractions of dimer products for the inhibitory effect of $NO_x$ as discussed above (Sect. 3.2.3). Within the uncertainty, the HOM yield at high NO in this study is comparable to the HOM yield (0.93±0.47)% from the photooxidation of limonene in a previous study (Jokinen et al., 2015). The difference in yield at low NO from (Jokinen et al., 2015) may be attributed to different experimental methods and conditions. For example, Jokinen et al. (2015) derived HOM yield from OH oxidation indirectly by comparing the HOM yield in ozonolysis with and without OH scavenger while in this study we determined HOM yield directly for the oxidation of limonene by OH. In ozonolysis, the cross reactions of

RO$_2$ formed from ozonolysis and OH oxidation may produce accretion products thus possibly confounding the results. By directly measuring HOM yield in the oxidation of limonene by OH, such influence is avoided. The NO concentration might be also different between our study and Jokinen et al. (2015). Additionally, the HOM molar yields related to H-abstraction were further determined to be $0.77^{+0.98}_{-0.41}$% at low NO and $0.10^{+0.13}_{-0.05}$% at high NO, when assuming that the ratio of all HOM from H-abstraction to those from OH addition is equal to the corresponding ratio for C$_{10}$-monomers.

## 4 Conclusions and atmospheric implications

In this study, the HOM products in the oxidation of limonene with OH were investigated by using a NO$_3^-$-CIMS in the SAPHIR chamber in Jülich. A large number of HOM monomers (C$_{6-10}$) and dimers (C$_{17-20}$) were detected and classified according to the number of carbon atoms. Among HOM products, the proportion of HOM monomers is far higher than dimers in both low and high NO condition. C$_{10}$ HOM were the most abundant monomers at both low and high NO. At low NO, the fraction of
organic nitrates (31%) was lower than that of non-nitrate HOM (69%). At high NO, a higher fraction of 41% were observed for organic nitrates. Two major RO$_2$ radical families, C$_{10}$H$_{15}$O$_x$• and C$_{10}$H$_{17}$O$_x$•, were identified. While C$_{10}$H$_{17}$O$_x$• were formed via OH addition to a double bond, C$_{10}$H$_{15}$O$_x$• is proposed to be formed via H-abstraction by OH on the basis of its molecular formula and the available literature. Although the concentrations of C$_{10}$H$_{15}$O$_x$• related products were less than C$_{10}$H$_{17}$O$_x$• related products, they remained at a comparable magnitude (38.9% vs 55.0% at low NO and 33.7% vs 49.7% at high NO). The
formation pathways of C$_{10}$H$_{15}$O$_x$• and C$_{10}$H$_{17}$O$_x$• were proposed on the basis of observed signal intensity, theoretical kinetic calculations, and structure activity relationships (SARs) for autoxidation. At both low and high NO, H-abstraction by OH contributes a significant fraction of HOM (41% and 42% of C$_{10}$ HOM monomer, respectively), demonstrating that H-abstraction is important to the formation of HOM in the oxidation of limonene with OH radical, and contributes more than expected given the ~34% contribution in the initiation reaction. The key mechanistic difference between OH addition and H-
abstraction pathways is that the former does not readily lead to breaking of the six-membered ring without chain termination, whereas H-abstraction leads to fast first-generation autoxidation after an alkoxy-peroxy ring breaking step. C$_6$-C$_9$ monomers are proposed to be formed via fragmentation of alkoxy radicals and C$_{17}$-C$_{20}$ dimers are proposed to be formed via accretion reactions.

    The molar yields of total HOM are estimated to be $1.97^{+2.52}_{-1.06}$% and $0.29^{+0.38}_{-0.16}$% in low and high NO condition. Although
the HOM yields in the oxidation of limonene with OH in this study are lower than that in the ozonolysis of limonene (Jokinen et al., 2015), the corresponding SOA mass yields assuming irreversible condensation of HOM can be as high as 4.3% and 0.6% at low NO and high NO, respectively. Similarly, while the gas-phase dimers observed in this study are clearly lower than monomers in abundance, dimers may also be very efficient at condensing onto newly formed particles due to their large size and low volatility. These HOM dimer products are known to have an extremely low volatility due to their size and high degree of oxidation (Tröstl et al., 2016). At 17 ppb NO the RO$_2$ fate is exclusively reaction with NO, and is representative for all
environment where RO$_2$ loss is dominated by the reaction with NO. Such environment includes urban regions and sub-urban regions, especially in developing countries such as East Asia and South Asia, where our HOM yield can be used to model these areas. In contrast, our HOM yield at low NO is representative for rural and remote continental environment (Rohrer et al., 1998; Lelieveld et al., 2008; Whalley et al., 2011; Moiseenko et al., 2021; Wei et al., 2019). Combined, our results can
thus be used directly in atmospheric chemical transport models to simulate the HOM concentrations in many atmospheric regimes (Pye et al., 2019; Xu et al., 2022) and help refine the simulations to assess its importance in new particle formation and particle growth (Zhao et al., 2020).

    This study highlights the importance of the pathway of H-abstraction by OH in competition to OH addition to double bonds in the same molecule at least for HOM formation, which has largely been neglected in current chemical mechanisms. The
importance of H-abstraction pathway to HOM formation also in α-pinene+OH oxidation has been shown in our recent study

(Shen et al., 2022), and recent experimental work by Williams et al. (2022). H-abstraction in the case of limonene+OH is more prominent than for α-pinene+OH, but the OH abstraction pathway is important in both systems suggesting its importance in other terpenes or even other unsaturated species. We propose that in order to accurately simulate HOM formation from oxidation of limonene by OH, H-abstraction should be considered in chemical mechanisms of atmospheric models.

Considering the key role HOM in SOA particle formation and growth, this study further enables more accurate simulation of chemical composition and concentrations of secondary organic aerosol as well as growth of particles to CCN size in order to assess the impact of SOA on climate.

The experiments in this study were conducted at ambient relevant conditions by using limonene and OH concentrations at ambient levels and using natural sunlight. The major $RO_2$ loss rate in all experiments is via $RO_2$+NO. Therefore, the HOM

composition and formation pathway can represent a large part of the daytime continental environment. The low NO conditions are representative of forested regions with biogenic monoterpene emissions and influenced by anthropogenic emissions and of rural regions. The experiments at high NO in this study are relevant for the reactions of limonene with OH in urban environment as limonene is also a major component of VCP besides biogenic sources (Nazaroff and Weschler, 2004; Rossignol et al., 2013; Waring, 2016; Gkatzelis et al., 2021).

The fraction of organic nitrate HOM, such as $C_{10}H_{15}NO_x$ and $C_{10}H_{17}NO_x$, was significant in the total HOM products in both low and high NO conditions examined, highlighting the importance of organic nitrates (ONs). Even at ~0.2 ppb NO, ONs account for a large part (31%) of HOM. ONs are important in the atmosphere as they can serve as an $NO_x$ reservoir (Ng et al., 2017). Highly functionalized ONs have been inferred to be capable of strongly partitioning to the particle phase (Perraud et al., 2012; Ng et al., 2007). The significant amounts of monoterpene-derived ONs that were observed in field campaigns

(Massoli et al., 2018; Huang et al., 2019; Lee et al., 2016) indicate that ONs make up a significant fraction of SOA. The abundant presence of ONs highlights the importance of $NO_x$-driven chemistry in daytime. In most of the continental environment influenced by monoterpene, ONs then likely contribute a large fraction of HOM and SOA, and contribute the organic nitrates and particle formation in the atmosphere.

**Acknowledgement**

H. Luo, H. Shen, and D. Zhao would like to thank the funding support of National Natural Science Foundation of China (No. 41875145), Science and Technology Commission of Shanghai Municipality (No. 20230711400), and Shanghai International Science and Technology Partnership Project (No. 21230780200). Sungah Kang, Astrid Kiendler-Scharr, and Thomas F. Mentel acknowledge the support by the EU Project FORCeS (grant agreement no. 821205).

**Data availability**

The supplement provides additional figures and tables, and further information on the CIMS calibration and HOM monomers and dimers with reduced carbon number.

The quantum chemical data (geometries, vibrational wavenumbers, rotational constants, energies, and partition functions) can be found under https://doi.org/10.26165/JUELICH-DATA/9JVHEK. [Reviewers: the quantum chemical data can be accessed

for reviewing purposes under URL https://data.fz-juelich.de/privateurl.xhtml?token=60754bd4-d921-449c-9210-7b24839a67ce]

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
