# Peer review of "Formation of highly oxygenated organic molecules from the oxidation of limonene by OH radical: significant contribution of Habstraction pathway"

_Atmospheric Chemistry and Physics, 2022_

## Referee Comment (RC1)

Review of Luo et al. ACP submission: Formation of highly oxygenated organic molecules from the oxidation of limonene by OH radical: significant contribution of H- abstraction pathway

**Significance:**

This is a well-motivated study about an important general process that is currently neglected in atmospheric chemistry models. The common chemistry models assume that an OH radical will always find its way to a double bond although several well-known facile H-abstractions would be available in the same molecule; the same double bond catalyzes the H-abstraction adjacent to it forming an allylic alkyl radical. This is quite well understood effect, but practically completely neglected in the current models describing atmospheric processes. This lack is likely most severe for secondary aerosol modelling, in which the condensing vapors are generated in-situ by rapid chain like oxidation chemistry, which likelihood is controlled by the oxidizing hydrocarbon structure. Only a fraction of the overall pathways leads to low volatile, condensable products, and thus describing the correct paths becomes critically important. Thus the study and analysis are timely and well-motivated.

While I find the topic worthwhile and certainly interesting for the readers of ACP, I am poorly convinced that such a parameter could be derived in a setup like SAPHIR. By reading the work through, I am sadly not much more convinced. Several crucial approximations have to be made to get started with the analysis, and with such a long processing times it becomes a tedious task to fill in the gap between the first reaction step (i.e., the reaction that is studied), and the chain like chemistry progressing to observed products through a complex and convoluted mechanism. I want to emphasize that the whole mechanism is littered with uncertainty, and with longer time scale this uncertainty only grows. Additionally, there seems to be some confusion with the presentation and some worrying observations about NOx have been made that put the final result in doubt. I'll detail my concerns below.

**Major comments:**

So first of all: Can you estimate such a quantity with this platform and experimental setup? Ideally this sort of work should be performed with techniques capable of seeing the primary radical products at short reaction times (e.g., resonance fluorescence, photoionization mass spectrometry, IR-spectrometry, etc.,), and not deduce the value from a very complex mechanism at very long reaction times (note that it seems the reaction time is not given in the manuscript text), with several assumptions about the mechanism and the detection. Such long residence times are arguably poorly suited to study details of chemical mechanisms, and are far more better equipped to study, for example, SOA forming potential. A large volume implies long time-scales, which leaves open the possibility for even slow processes to make a dent. The bare minimum is that the caveats of this approach should be discussed.

Several details seem to be missing which prevent understanding how the work was performed and analyzed. First of all, what was the OH source? You are reporting OH oxidation experiments but it appears you do not even mention the OH source in the main text. The method of OH production should be discussed extensively especially if you claim the current work was done better than the previous. Also, what was the source of HO2? How was it controlled? Was it? What were the used limonene concentrations? What was the time-scale

of the experiment? Where is the HONO originating from? Does it prevent making such a study as apparently the chamber has always appreciable NOx present. To clarify, it is really difficult to put the results into any context when the most important parameters are left out. I hope you remember that the study should reproducible with the given information, and that the ACP format does not require for a shorter length. Also, please remember that the supplemental material is for adding information that is not pertinent for understanding the work, but is a place to add additional information supporting the claims.

How well does the "HOM measured under 17 ppb of NOx in a one chamber setup" represent the general "high NOx" yield of limonene HOM? Is this realistic to use for atmospheric modelling? Would some sort of fit between the conditions be better, or is it sufficient to use a one value for all the environments?

On the same lines: What does "a HOM yield of a single molecule" actually signify? Can you apply it for an air quality study, or how is it a useful quantity? How much does it depend on the individual configuration of the experiment and the instruments? When you consider that individual HOM measurement is affected by the transmission and fragmentation of the instrument, the detection sensitivity due to changes in diffusion of bulky HOM, detection (=charging) probability that depends on the exact molecular structure, and so on... So to sum it up: how realistic is any given HOM yield, and how useful it is for others to utilize? This should be discussed.

The given range for "low NO" spans over 3 orders of magnitude(!), whereas the "high NO" is only about 80 times higher than highest low. This seems a bit strange. Even more strange is that you obtain 33% yield for the nitrates even with the low NO conditions – to me it would imply that your low and high conditions were far closer that what you assume here. Also it raises the question that is it possible to deduce the OH reaction influence with such a high persistent background levels of NOx? A sensitivity analysis based on the model chemistry would improve the credibility.

I generally find the discussion around this NO influence confusing. For example you state that "to estimate the impact of ozone oxidation during photoxidation, we calculated the reaction rates of VOC+ OH and VOC+O3 in low and high NO conditions of this study.." without explaining where this difference comes from. Obviously the NO has nothing to do with the VOC + OH and VOC + O3 rates unless you consider the secondary chemistry. This can be very misleading to a reader without a prior good grasp on the ongoing processes. Please connect these statements to the chemical phenomenon you're describing.

**To consider:**

*Do you consider the RO2 + HO2 $\rightarrow$ RO + O2 + OH pathway for any of the RO2 in the mechanism? This should be possible as you follow the structures explicitly.

*Why can't the C10H17Ox produce an C10H16Oy alcohol by RO2 + RO2?

*Is it really the case you saw no trimer species? By looking at the Figure 1, it appears that dimers are high, and four a two double bond systems trimers have been reported (e.g.,

Molteni et al. https://acp.copernicus.org/articles/18/1909/2018/)). So if you would provide me a Zoom of the 700-900 Th – are there no apparent bands of peaks present?

*The method (Fuchs et al. 2012) used to determine [RO2] and [HO2] has been debated to be prone to artefacts from different RO2 propagation reactions. For example, a tertiary RO2 will not lead into HO2 and is thus miscounted, especially here when your most facile reaction advances through a tertiary RO2. Was any correction applied to the values of [RO2] and [HO2]? If not, the caveat of not doing so should be the very least discussed.

*Line 348: Yes, the alkoxy-peroxy step removes the geometrical constraint, but it also imposes a limitation on the efficiency of this path, as the bimolecular step is needed to allow for autoxidation, whereas losses by wet and dry scavenging, photolysis and reactions with other trace gases might as well happen. Postulating several alkoxy-peroxy sequences will impose even greater limitations, as the radicals may be lost in each step and the OH is mainly lost with the high [VOC]. This caveat on the way to HOM should be clearly stated.

*How good is the assumption to use "the same $k_{uni}$, $k_{RO2+RO2}$, $k_{RO2+HO2}$, and α for a given $C_{10}H_{15}O_x\bullet$ and $C_{10}H_{17}O_x\bullet$ family."? Easy examples where this does not hold are: H-shift across the ring vs H-shift in a linear chain after ring breaking. RO2 + RO2 for a primary RO2 vs oxygenated RO2. How big of an uncertainties these assumption generate?

*How does the shifting detection sensitivity of CIMS affect your conclusions (e.g. Hyttinen et al. https://pubs.acs.org/doi/10.1021/acs.jpca.7b10015), as the $C_{10}H_{17}O_x$ is likely detected better than the $C_{10}H_{15}O_x$ products, especially at the lower end of the oxygen content?

*It seems unlikely that the SAR will reproduce the rates of the highly oxygenated molecule H-shift rates. This is due to intramolecular interactions (mainly H-bonding) that influences the thermochemistry; a favorable interaction on the reactant side will increase the barrier to reaction, and can significantly decrease the H-shift rate. Looking at the schemes of the paper with the given SAR rates can leave a very wrong picture for the reader. This influence is hinted in the manuscript but, still the uncertain numbers are presented in the schemes. Is this reasonable?

*The secondary OH reaction is generally inherently less likely in lab systems, as I would expect to be here as well (i.e., how much would you need to accumulate the product before it'll find the second OH at a rate sufficient for measurable product formation?). Yet, with the missing documentation about the used concentration ranges it is not possible to even roughly estimate this fraction.

*Line 323: "However, this extra bimolecular OH reaction takes time, and can delay formation of HOM. An analysis of the secondary chemistry is outside the scope of this work, …" As mentioned above, not all pathways lead to HOM, and the whole paper is an analysis of secondary chemistry if you are studying OH abstraction vs OH addition through a complex mechanism. Please sharpen your words and thinking.

*Line 328: Here you imply that the low NO is actually the highest of your reported range (0.2 ppb) and quote a value of 29% for RONO2 which is not in line with the previously given 33%.

At these conditions this seems like a very high value. Have you considered, for example, that your CIMS could be more sensitive to nitrates than to HOM devoid of -ONO2? There are some hints in the recent literature about this.

*Finally, it is very difficult for me to see why higher NO can lead to higher abstraction vs addition rates. These issues seem uncoupled (i.e., no amount NO can increase an OH abstraction rate) and rather point into being misunderstood RO2 + NO (or indeed RO + NO) chemistry connected to the long processing time-scales.

**Minor comments:**

*Consider changing the "dimer" into "accretion product".
*Line 66: "Biomolecular" is an error. (Other places as well).
*Line 150: What "organic" are you referring to? The identity matters a lot, as has been shown in this chamber setup previously.
*Line 158: delta symbol is usually reserved for a changing quantity.
*Line 210: How should one understand the statement: "Briefly, we only consider a limited oxidation network by considering all possible reaction channels, …"
*Don't mix -y and -yl endings, i.e., if you choose to use "peroxyl" then use "alkoxyl" as well.
*Line 394: Didn't catch why you presume the disproportionation of RO2 + RO2 would favor carbonyls over alcohols.
*Plural of formula is formulae.

---

## Referee Comment (RC2)

**Overall comment:**

This work by Luo et al. examined HOM formation from limonene + OH and highlighted the importance of the H-abstraction pathways, in addition to the well-known OH-addition pathways. The main conclusion is well supported by the observations of C10H15Ox-RO2 as well as their termination products. These results are also in line with the authors' recent publication on H-abstraction derived HOM for alpha-pinene + OH chemistry (Shen et al., 2022 Sci. Adv.). Therefore, I believe that the underscored H-abstraction pathways for monoterpene + OH chemistry are important and should be considered in future chemical mechanisms. However, there are a number of major issues that need to be addressed before consideration for publication, as discussed below.

**Detailed comments:**

1. Estimation of HOM yields. This work used NO3-CIMS to estimate HOM yields and relative contributions of the H-abstraction vs. OH-addition pathways, assuming unified sensitivity for all the HOMs products. However, this approach may have large uncertainties. For example, Berndt et al. (Nature Comm. 2016, 7, 13677) demonstrated that NO3-CIMS could underestimate HOM formation from OH-addition pathways for alpha-pinene and beta-pinene. This might also be the case for limonene. If this is true, then the contribution of OH-addition to limonene HOMs as well as the total HOM yields could be underestimated. Better constraints on these aspects are needed.

2. RO2 chemistry. In Page 2, the RO2 reactions are listed. However, there are a few missing pathways. First, the RO2 + HO2 reaction may only partially form ROOH and could also produce RO (Kurten et al., JPCL, 2017, 8, 2826). The branching ratios of the two pathways likely depend on the RO2 structure. In addition, RO unimolecular isomerization could produce RO2 in the same CHO family. For example, C10H15Ox RO could form C10H15O(x+2) RO2. Therefore, this pathway needs to be included in the data analysis. Lastly, the RO2 + RO2 reaction described in (R2) is only correct for primary and secondary RO2. For tertiary RO2s, carbonyls cannot be formed. Without considering these above-mentioned pathways, the estimation of contributions from C10H15Ox vs. C10H17Ox to C10H16Ox (i.e., Eq. 2 - Eq. 8) is wrong.

3. Products in the low-NO conditions. A range of NO concentration was reported (0.1 ppt – 0.2 ppb) for the low-NO conditions. So, I assume there were multiple experiments performed under low-NO conditions with varied NO concentrations from as low as 0.1 ppt up to 0.2 ppb. Then, I would expect the product distributions under varied NO concentrations to be reported. However, in Figures 2 and 3, only one set of results were shown for low-NO. What is the NO concentration for the low-NO conditions shown? It is even more confusing that the fractions of C6-C9 fragmentation products (Figure 2) and organic nitrates (Figure 3) are not significantly lower for the low-NO experiments in comparison to the high-NO experiments. Even at 0.2 ppb for low-NO, it is ~ 100 times lower than the high-NO condition. But the fragmentation products are only different by 7%, and organic nitrates are 34% vs. 73%. More thorough analysis is needed to verify the difference.

4. Product distributions. In Figure 3, what is there a strong NOx dependence for the C10H15Ox/C10H17Ox ratio? Why are the ratios of C10H18Ox/C10H17Ox fairly constant (28% vs.

19%) between the two NO conditions, but C10H16Ox(R-OH/OOH)/C10H15Ox ratios are very different (51% vs. 0%)? The relationships between different families under different NO conditions need to be discussed in-depth.

5. The authors suggest that some C10H17Ox and related products may be from secondarygeneration reactions. For example, the first-generation product limonaldehyde (C10H16O2) could react with OH and form C10H17Ox. In the same way, H-abstraction in limonaldehyde reacting with OH to form C10H15Ox RO2 and further products. In MCM, H-abstraction in this reaction has a branching ratio of ~28.8%. Could this reaction explain some or a major fraction of total C10H15Ox products?

Minor comments:

1. The concentration of limonene used in the experiments are not mentioned in the main text.

2. Suggested references:Line 72. For HOM dimers: Zhao et al. PNAS, 2018, 115, 12142.Line 89. For limonene + NO3: Mayorga et al., ES&T, 2022, 56, 15337.

3. Line 84. Should be C10H14-18O7-11.

4. Line 93. Here the actual HOM yield from Jokinen et al. should be mentioned.

5. NOx analyzer. What is the detection limit of the NOx analyzer? The manuscript says low-NO has NO concentration of 0.1 ppt – 0.2 ppb but did not clearly say that the NOx analyzer measured the NO concentration.

6. Line 146-147. This sentence should move to the results section.

7. Line 163. How are epoxides formed?

8. In the equations, sometimes R-H=O is used for carbonyls and sometimes R=O is used. They need to be the same.

9. Line 394. Carbonyls may also be formed from RO + O2 if the RO is primary or secondary.

10. Line 503. Molar yields? Line 505. SOA mass yields?

11. Line 522. Significant contribution of RO2 + HO2 after 2 h at low NO. But this study only focused on within 1 h after oxidation, correct?

---

## Author Comment (AC1)

We thank the reviewer for the helpful comments on our manuscript. The comments are greatly appreciated. We have addressed all the comments and believe that the revisions based on the comments help improve the quality of our manuscript. Below please find our responses to the comments one by one and the corresponding revisions made to the manuscript. The original comments are in italics. The revised

5     parts of the manuscript are in blue.

*Review of Luo et al. ACP submission: Formation of highly oxygenated organic molecules from the oxidation of limonene by OH radical: significant contribution of H- abstraction pathway*
***Significance:***

10   *This is a well-motivated study about an important general process that is currently neglected in atmospheric chemistry models. The common chemistry models assume that an OH radical will always find its way to a double bond although several well-known facile H-abstractions would be available in the same molecule; the same double bond catalyzes the H-abstraction adjacent to it forming an allylic alkyl radical. This is quite well understood effect, but practically completely neglected in the current*

15   *models describing atmospheric processes. This lack is likely most severe for secondary aerosol modelling, in which the condensing vapors are generated in-situ by rapid chain like oxidation chemistry, which likelihood is controlled by the oxidizing hydrocarbon structure. Only a fraction of the overall pathways leads to low volatile, condensable products, and thus describing the correct paths becomes critically important. Thus the study and analysis are timely and well-motivated.*

20

*While I find the topic worthwhile and certainly interesting for the readers of ACP, I am poorly convinced that such a parameter could be derived in a setup like SAPHIR. By reading the work through, I am sadly not much more convinced. Several crucial approximations have to be made to get started with the analysis, and with such a long processing times it becomes a tedious task to fill in the gap between the*

25   *first reaction step (i.e., the reaction that is studied), and the chain like chemistry progressing to observed products through a complex and convoluted mechanism. I want to emphasize that the whole mechanism is littered with uncertainty, and with longer time scale this uncertainty only grows. Additionally, there seems to be some confusion with the presentation and some worrying observations about NOx have been made that put the final result in doubt. I'll detail my concerns below.*

30

***Major comments:***
*So first of all: Can you estimate such a quantity with this platform and experimental setup? Ideally this sort of work should be performed with techniques capable of seeing the primary radical products at short reaction times (e.g., resonance fluorescence, photoionization mass spectrometry, IR-spectrometry, etc.,),*

35   *and not deduce the value from a very complex mechanism at very long reaction times (note that it seems the reaction time is not given in the manuscript text), with several assumptions about the mechanism and the detection. Such long residence times are arguably poorly suited to study details of chemical mechanisms, and are far more better equipped to study, for example, SOA forming potential. A large volume implies long time-scales, which leaves open the possibility for even slow processes to make a*

40   *dent. The bare minimum is that the caveats of this approach should be discussed.*
**Response:**
We assume that the reviewer mentioned "quantity" refers to the importance of the OH H-abstraction in HOM formation. We agree that some slow processes may confound the deduction of the importance of OH H-abstraction over long reaction time, for example, the second-generation reactions, and prior to the

45 experiments we did not anticipate that this aspect would be so important and hence did not particularly design for it. However, as seen in Figure 4, the dominance in the HOM formation of $C_{10}H_{15}O_x\bullet$ formed from H-abstraction is clear already in the very first few minutes of the experiments, where long term effects or chamber influences are virtually absent. Specifically, the large volume guarantees that the influence of wall reactions is negligible relative to the chemistry time scales as determined by the reactant

50 concentrations. While the time length was 60 min in our original manuscript, in the revised manuscript we focus on the early stage of reaction (the first 15 min) in order to minimize the possible contribution from the second-generation reactions. Similarly, in this early stage, second-generation chemistry of OH with the first-generation products is negligible compared to the reaction of limonene (Fig. S9). This can be quantified using the following comparison of the respective reaction rate of OH via hydrogen

55 abstraction with limonene and limonaldehyde, which is the dominant first-generation $C_{10}$ product (99% among $C_{10}$ products). We calculated the relative reaction rate of hydrogen abstraction from limonene to that from limonaldehyde by OH radicals, as shown below:

$$\frac{R[LIM + OH]_{H\ abstraction}}{R[LIMAL + OH]_{H\ abstraction}} = \frac{k[LIM + OH] \times [LIM] \times [OH] \times BR_{LIM}[H\ abstraction]}{k[LIMAL + OH] \times [LIMAL] \times [OH] \times BR_{LIMAL}[H\ abstraction]}$$

$$= \frac{k[LIM+OH] \times [LIM] \times BR_{LIM}[H\ abstraction]}{k[LIMAL+OH] \times [LIM]_r \times Y[LIMAL] \times BR_{LIMAL}[H\ abstraction]} \qquad \text{Eq. S4}$$

60 where k[LIM+OH] and k[LIMAL+OH] are reaction rate constants (Atkinson, 1997). Here, $BR_{LIM}$[H-abstraction] and $BR_{LIMAL}$[H-abstraction] are the branching ratios for limonene + OH and limonaldehyde + OH reactions to undergo H-abstraction channel, respectively. A branching ratio of 0.34 for limonene + OH was used (Rio et al., 2010) and a ratio of 0.29 for limonaldehyde + OH was used based on MCM v3.3.1 (http://mcm.york.ac.uk/). The concentrations of limonene were directly measured while

65 concentrations of limonaldehyde were estimated according to their NO-dependent yields (Y[LIMAL] in Equation 1), with a value of 0.29 at low NO and 0.28 at high NO (Hakola et al., 1994). The uncertainties

of $\frac{R[LIM+OH]_{H\ abstraction}}{R[LIMAL+OH]_{H\ abstraction}}$ was estimated to be -41%/+141% at low NO and high NO, from the uncertainty

of limonene concentration (~15%), k[LIM+OH] ($\Delta$logk=$\pm$0.08), and Y[LIMAL] ($\pm$0.06 at low NO and high NO (Hakola et al., 1994)) using error propagation. Hydrogen abstraction from limonene is 19-1600

70 times faster than that from limonaldehyde at low NO and 29-87 times faster at high NO. Note that the concentrations of limonaldehyde were estimated from consumed limonene and yields of limonaldehyde, thus reflecting only the production. As limonaldehyde is continuously consumed by OH, its true concentration should be lower, and thus its relative importance is even overestimated using this method.

Moreover, we ran MCM model with H abstraction of OH from limonene and obtained similar results

75 of the relative reaction rates of OH abstraction from limonene and from limonaldehyde (Figure S9).

Overall, we thus conclude that even the dominant product limonaldehyde likely has only a negligible contribution to HOM formation at early stages of the experiments. Therefore, second-generation reactions are unlikely to contribute the $C_{10}H_{15}O_x$-related HOM observed in our study at those time scales.

In the revised manuscript, we added the reaction time in line 121 in the revised main text:

80 "To avoid possible interference due to long reaction time, the subsequent discussion focuses on the early stage (15 min) of the experiment. The initial experimental conditions are shown in Table S1."

And expended the discussion in line 379 in the revised main text:

"In principle, $C_{10}H_{15}O_x\bullet$ peroxy radicals might also formed through secondary chemistry of first-generation $C_{10}$ oxidation products of the limonene+OH reaction. The limonaldehyde ($C_{10}H_{16}O_2$) is the

85 most abundant (99%) first-generation $C_{10}$ product reported in limonene+OH reaction (Hakola et al., 1994;

Larsen et al., 2001), which can form $C_{10}H_{15}O_4\bullet$ and the $C_{10}H_{15}O_x\bullet$ family by further autoxidation through H-abstraction and subsequent $O_2$ addition. Therefore, we take limonaldehyde into account as the most competitive candidate. For the early stages of our experiments (first 15 min), however, we find that secondary chemistry is not important (Section S2 and Fig. S9 in Supplement)."

90    We added the comparison of limonene+OH and limonaldehyde+OH in the ==Section S2== and ==Figure S9== in the revised supplement:

"The importance of secondary chemistry is not important in this studythe C10H15Ox-related HOM formation. This can be demonstrated by the following comparison of the chemistry of the limonene and limonaldehyde, which is the dominant first-generation $C_{10}$ product (>99%). To quantify the relative

95    importance of these two pathways, the relative reaction rates of hydrogen abstraction from limonene+OH to that from limonaldehyde+OH were calculated as below:

$$\frac{R[LIM+OH]_{H\ abstraction}}{R[LIMAL+OH]_{H\ abstraction}} = \frac{k[LIM+OH] \times [LIM] \times [OH] \times BR_{LIM}[H\ abstraction]}{k[LIMAL+OH] \times [LIMAL] \times [OH] \times BR_{LIMAL}[H\ abstraction]}$$

$$= \frac{k[LIM+OH] \times [LIM] \times BR_{LIM}[H\ abstraction]}{k[LIMAL+OH] \times [LIM]_r \times Y[LIMAL] \times BR_{LIMAL}[H\ abstraction]} \qquad \text{(Eq. S4)}$$

where k[LIM+OH] and k[LIMAL+OH] are reaction rate constants based on MCM v3.3.1 (Atkinson,

100    1997). [LIM], [LIMAL], and [OH] are the concentrations of limonene, limonaldehyde, and OH radicals, while limonene and OH radicals concentrations were measured and concentrations of limonaldehyde were estimated on the basis of their NO-dependent yields (0.29 at low NO and 0.28 at high NO) (Y[LIMAL] in Equation S4) (Hakola et al., 1994). $BR_{LIM}$[H abstraction] and $BR_{LIMAL}$[H abstraction] are the branching ratio of H-abstraction channel from limonene + OH and limonaldehyde + OH, respectively.

105    The branching ratio is 0.34 for the reaction limonene + OH (Rio et al., 2010) and 0.29 for limonaldehyde + OH based on MCM v3.3.1 (http://mcm.york.ac.uk/). The uncertainties of the relative reaction rates were estimated to be -41%/+141% at low NO and high NO, from the uncertainty of limonene concentration (~15%), k[LIM+OH] ($\Delta$logk=±0.08), and Y[LIMAL] (±0.06 at low NO and high NO) using error propagation. As a result, hydrogen abstraction from limonene is 19-1600 times faster than

110    that from limonaldehyde at low NO and 29-87 times at high NO (Fig. S9). Note that the concentrations of limonaldehyde were estimated from consumed limonene, which only reflect the production and neglect consumption. Therefore, the relative importance of limonaldehyde was even overestimated using this method. Based on this evidence, the contribution of limonaldehyde to HOM formation was likely negligible at early stages of the experiments. Therefore, the second-generation reactions are unlikely to

115    contribute the $C_{10}H_{15}O_x$-related HOM observed in our study."

[Figure]

**Figure S9.** The relative ratio of hydrogen abstraction rate of the reaction limonene+OH to that of the reaction limonaldehyde+OH within the first 15 min reaction time obtained from measured at low NO (a, c) and high NO (b, d). Panels a-b and c-d show the results obtained from measured limonene concentration and limonaldehyde yield and from MCM modeling, respectively. The dashed lines are at the value of 10 (i.e., ~10% contribution of secondary chemistry). Note that different scales of y axes between panel (a, c) and (b, d). The large change in panel (b) results from the large measurement uncertainty of low accumulated limonene consumption measured by PTR-ToF-MS in the first few minutes.

*Several details seem to be missing which prevent understanding how the work was performed and analyzed. First of all, what was the OH source? You are reporting OH oxidation experiments but it appears you do not even mention the OH source in the main text. The method of OH production should be discussed extensively especially if you claim the current work was done better than the previous. Also, what was the source of HO2? How was it controlled? Was it? What were the used limonene concentrations? What was the time-scale of the experiment? Where is the HONO originating from? Does it prevent making such a study as apparently the chamber has always appreciable NOx present. To clarify, it is really difficult to put the results into any context when the most important parameters are left out. I hope you remember that the study should reproducible with the given information, and that the ACP format does not require for a shorter length. Also, please remember that the supplemental material is for adding information that is not pertinent for understanding the work, but is a place to add additional information supporting the claims.*

**Response:**

We thank the reviewer for the helpful comment. We clarify these details below and in the revised manuscript.

Regarding OH source, in low and high NO experiments, OH radicals were generated from the photolysis of HONO. HONO were produced from a well-characterized photolytic source related to the Teflon wall, which we have mentioned in line 138 of our original manuscript. Besides this OH source, OH was further formed via the photolysis of $O_3$ formed during the photo-oxidation of limonene.

Regarding $HO_2$ sources, $HO_2$ was produced during the photo-oxidation of limonene, i.e., $RO_2+NO \rightarrow RO+NO$ and the reaction of RO with $O_2$ during the experiments. There is no extra $HO_2$ source,

and we did not control its concentration in our experiments.

The concentrations of limonene are 7 ppb. The initial experimental conditions are now shown in Table S1. We focus on the first 15 minutes in this study, although the duration of the experiments lasts for several hours. Meanwhile, the time resolution of the CIMS instrument is 4 seconds.

We added a detailed description to the revised main text in line 140:

"OH radicals were generated from the photolysis of HONO in both low and high NO experiments and the HONO was formed from the Teflon chamber wall via a photolytic process. The details have been described by Rohrer et al. (2005). $HO_2$ was produced from the reaction of $O_2$ with RO, which can be formed in the reaction of $RO_2$+NO in photo-oxidation during the experiments. The concentration of limonene was 7 ppb. The reaction time after the roof opened was 8 hours."

*How well does the "HOM measured under 17 ppb of NOx in a one chamber setup" represent the general "high NOx" yield of limonene HOM? Is this realistic to use for atmospheric modelling? Would some sort of fit between the conditions be better, or is it sufficient to use a one value for all the environments?*

**Response:**

At 17 ppb NO the $RO_2$ fate is exclusively dominated by its reaction with NO. The HOM yield at 17 ppb NO represents the environment where $RO_2$ mainly react with NO. Such environments include urban regions and sub-urban regions, especially in developing countries such as East Asia and South Asia. Therefore, it is realistic to use the HOM yield to model these areas.

It is certainly not sufficient to use one value for all environment. Here we also measured the HOM yield at low NO to represent the rural and remote continental environment (Rohrer et al., 1998; Lelieveld et al., 2008; Whalley et al., 2011; Moiseenko et al., 2021; Wei et al., 2019).

In the revised manuscript, we discussed the atmospheric relevance of the HOM yield obtained in this study as follows and extend the discussion in line 589 in the revised main text:

"At 17 ppb NO the $RO_2$ fate is exclusively reaction with NO, and is representative for all environment where $RO_2$ loss is dominated by the reaction with NO. Such environment includes urban regions and sub-urban regions, especially in developing countries such as East Asia and South Asia, where our HOM yield can be used to model these areas. In contrast, our HOM yield at low NO is representative for rural and remote continental environment (Rohrer et al., 1998; Lelieveld et al., 2008; Whalley et al., 2011; Moiseenko et al., 2021; Wei et al., 2019)."

*On the same lines: What does "a HOM yield of a single molecule" actually signify? Can you apply it for an air quality study, or how is it a useful quantity? How much does it depend on the individual configuration of the experiment and the instruments? When you consider that individual HOM measurement is affected by the transmission and fragmentation of the instrument, the detection sensitivity due to changes in diffusion of bulky HOM, detection (=charging) probability that depends on the exact molecular structure, and so on… So to sum it up: how realistic is any given HOM yield, and how useful it is for others to utilize? This should be discussed.*

**Response:**

The HOM yield refers to a class of compounds, not just a single molecule. HOM have low to extremely low volatility due to its high oxygenation and exists in the gas phase, and have an important contribution to SOA. Obtaining HOM yields allows one to estimate SOA yield contributed by HOM. It can be used in atmospheric chemical transport models to simulate the HOM concentrations (Pye et al., 2019; Xu et al., 2022) and assess its importance in new particle formation and particle growth (Zhao et al., 2020). If

the reactions to form HOM and their contribution to SOA formation are incorporated into models, the accuracy of SOA concentration simulation in models can be further improved.

In the revised manuscript, we discussed the atmospheric relevance of the HOM yield as follows and extend the discussion in line 593 in the main text:

"Combined, our results can thus be used directly in atmospheric chemical transport models to simulate the HOM concentrations in many atmospheric regimes (Pye et al., 2019; Xu et al., 2022) and help refine the simulations to assess its importance in new particle formation and particle growth (Zhao et al., 2020)."

*The given range for "low NO" spans over 3 orders of magnitude(!), whereas the "high NO" is only about 80 times higher than highest low. This seems a bit strange. Even more strange is that you obtain 33% yield for the nitrates even with the low NO conditions – to me it would imply that your low and high conditions were far closer that what you assume here. Also it raises the question that is it possible to deduce the OH reaction influence with such a high persistent background levels of NOx? A sensitivity analysis based on the model chemistry would improve the credibility.*

**Response:**

The range of NO concentration is a typo. We apologize for this error. The range of NO at low NO was 0.06 - 0.1 ppb in the first 15 min. In our study, the difference of the low and high NO concentrations is about two orders of magnitude. As mentioned above, low and high NO concentrations represent rural and most remote continental regions and urban regions, respectively. We would like to note that in our study, low NO mostly referred to the clean continental environment with low NO concentrations of 0.1 - ~0.2 ppb (Rohrer et al., 1998; Lelieveld et al., 2008; Whalley et al., 2011; Moiseenko et al., 2021; Wei et al., 2019) and does not refer to the case where $RO_2$ loss is dominated by the reaction with $HO_2$, e.g., in remote oceanic environment. In our study at low NO the dominant $RO_2$ loss reaction is still $RO_2+NO$, which has been stated clearly in our original manuscript and noted again in the revised manuscript (Figure S3). This explains why there is considerable formation of organic nitrate under low NO conditions.

The $NO_x$ level does not affect how we deduce the contribution of OH abstraction in HOM formation as we have clearly specified the $NO_x$ level. We are not sure why a sensitivity analysis is necessary here. We have revised in line 139 in the main text:

"0.06 - 0.1 ppb"

And we have noted the meaning of low NO as follows in line 147 in the main text:

"We would like to note that the low NO does not refer to the case where $RO_2$ loss is dominated by the reaction with $HO_2$, e.g., in remote ocean environment."

*I generally find the discussion around this NO influence confusing. For example you state that "to estimate the impact of ozone oxidation during photoxidation, we calculated the reaction rates of VOC+ OH and VOC+O3 in low and high NO conditions of this study.." without explaining where this difference comes from. Obviously the NO has nothing to do with the VOC + OH and VOC + O3 rates unless you consider the secondary chemistry. This can be very misleading to a reader without a prior good grasp on the ongoing processes. Please connect these statements to the chemical phenomenon you're describing.*

**Response:**

We did not intent to indicate that NO influences VOC+OH or VOC+ $O_3$. We would like to state that in both conditions OH oxidation is the dominant reaction pathway of VOC. In our original manuscript, the

slightly higher contribution of $O_3$ at high NO is attributed to higher $O_3$ formed in the photo-oxidation of limonene compared to low NO. In the revised manuscript, as we only focus on the first 15 min, the difference is not noticeable (Figure S2 in the revised Supplement).

In the revised manuscript, we have clarified the point as follows and revised in line 145 in the main text:

"To estimate the impact of ozone oxidation during photo-oxidation, we calculated the reaction rates of VOC+ OH and VOC+$O_3$ in the experiments of this study (Fig. S2)."

***To consider:***

*\*Do you consider the RO2 + HO2 → RO + O2 + OH pathway for any of the RO2 in the mechanism? This should be possible as you follow the structures explicitly.*

**Response:**

This pathway is possible in principle. However, according to the $RO_2$ loss in Figure S3, $RO_2$ + $HO_2$ is not important at both low and high NO (less than 1% proportion of $RO_2$ loss) in our study.

In the revised manuscript, we added this reaction in the introduction in line 79.

*\*Why can't the C10H17Ox produce an C10H16Oy alcohol by RO2 + RO2?*

**Response:**

By $RO_2$ + $RO_2$ pathway, $C_{10}H_{17}O_x$ produce $C_{10}H_{18}O_x$ alcohol and $C_{10}H_{16}O_x$ carbonyl (Barbara J. Finlayson-Pitts).

*\*Is it really the case you saw no trimer species? By looking at the Figure 1, it appears that dimers are high, and four a two double bond systems trimers have been reported (e.g., Molteni et al. https://acp.copernicus.org/articles/18/1909/2018/)). So if you would provide me a Zoom of the 700-900 Th – are there no apparent bands of peaks present?*

**Response:**

We did not observe trimers in our limonene+OH system. We would like to note that the results of Molteni et al. (2018) were observed in the oxidation of aromatic hydrocarbons. The oxidation pathways and products can be quite different from that of limonene+OH in this study. We have added a zoom of 520-920 Th (Figure S4) in the revised Supplement.

[Figure]

**Figure S4.** The average mass spectrum in the range of 520-920 Th at low and high NO over the first 15 min.

*\*The method (Fuchs et al. 2012) used to determine [RO2] and [HO2] has been debated to be prone to artefacts from different RO2 propagation reactions. For example, a tertiary RO2 will not lead into HO2 and is thus miscounted, especially here when your most facile reaction advances through a tertiary RO2.*

*Was any correction applied to the values of [RO2] and [HO2]? If not, the caveat of not doing so should be the very least discussed.*

**Response:**

Concerning $HO_2$, we assume that the reviewer refers to the work by Fuchs et al. (2011), in which the potential artefact in $HO_2$ measurements from the concurrent chemical conversion of $RO_2$ in the instrument making use of chemical conversion of $HO_2$ by the reaction with NO is described. As shown in that work, the artefact can be avoided, if only a small NO concentration used. This was done in all experiments in this study, so that no corrections of $HO_2$ concentration measurements are required. Therefore, the method to determine $HO_2$ concentration is not affected by the artefact.

RO$_2$ radicals are detected as sum of $HO_2+RO_2$ and the contribution from $HO_2$ radicals is subtracted from the signal to derive $RO_2$ radical concentrations. The method relies on the conversion of $RO_2$ to $HO_2$ in their reaction with NO that is achieved in a conversion reactor of the instrument. The reviewer is correct that $RO_2$ species that do not yield $HO_2$ cannot be detected. For example, Novelli et al. (2021) found that the decomposition of beta-nitrate alkoxy radicals (formed in the reaction of nitrate $RO_2+NO$) form $NO_2$ instead of $HO_2$. However, such a reaction is only found for nitrate $RO_2$ up to now. Even tertiary $RO_2$ can still form $HO_2$ in the reaction with NO via the further decomposition of RO forming a new primary or secondary $RO_2$ (Ziemann and Atkinson, 2012; J. et al., 1997). In photochemistry experiments, there is no significant fraction of nitrate $RO_2$ formed. In this study, we do not expect that there were a large contribution of nitrate $RO_2$ to the sum of all $RO_2$. The large contribution of nitrate $RO_2$ would only be the case in experiments, in which the VOC is mainly oxidized by $NO_3$. We only make use of the $RO_2$ measurements to calculate the lifetime of $RO_2$ with respect to the different reaction channels. Therefore, this should not matter for the $RO_2$ lifetime calculation in photochemistry experiments, when $RO_2$ is mainly from the OH oxidation.

We did apply a correction to $RO_2$ concentrations. To make it clear, we added some description in line 128 in the revised main text:

"Note that the potential artefact in $HO_2$ measurements from the concurrent chemical conversion of $RO_2$ in instrument making use of chemical conversion of $HO_2$ by the reaction with NO can be avoided in this study through NO used, so that no corrections of $HO_2$ concentration measurements are required. The detection of $RO_2$ radicals relies on the conversion of $RO_2$ to $HO_2$ in their reactions with NO. We applied a correction to $RO_2$ concentrations (Fuchs et al., 2011). We would like to note that only a few nitrated $RO_2$ were observed to not form $HO_2$ in the reaction with NO. In this study, we do not expect that there were a large contribution of nitrate $RO_2$ to the sum of all $RO_2$ as in photochemistry experiments, as there is no significant fraction of nitrate $RO_2$ formed."

*\*Line 348: Yes, the alkoxy-peroxy step removes the geometrical constraint, but it also imposes a limitation on the efficiency of this path, as the bimolecular step is needed to allow for autoxidation, whereas losses by wet and dry scavenging, photolysis and reactions with other trace gases might as well happen. Postulating several alkoxy-peroxy sequences will impose even greater limitations, as the radicals may be lost in each step and the OH is mainly lost with the high [VOC]. This caveat on the way to HOM should be clearly stated.*

**Response:**

Losses of the $RO_2$ radicals by scavenging, photolysis and other trace gasses also occurs during HOM formation that does not proceed through an alkoxy-peroxy step, and thus are not an unusual limitation for autoxidation efficiency; the limonene alkoxy radicals are too short-lived to be affected by these

315 processes. During the experiments, the bimolecular step $RO_2+NO$ was fast (~0.01 s$^{-1}$ at low NO and 4 s$^{-1}$ at high NO, respectively), and was the dominant loss of the cyclic $RO_2$ in our study, such that the alkoxy-peroxy step is not limiting under those conditions. In the atmosphere, HOM yields always depend on the reaction conditions due to competition, but for limonene the yields may be peak at higher NO concentrations than for non-cyclic VOCs due to the alkoxy-peroxy step. However, neither the alkoxy-

320 peroxy step or fragmentation of the alkoxy steps breaks the radical chain, and autoxidation can proceed without further OH reactions.

We have extended this section in the revised manuscript, emphasizing that the need for one alkoxy-peroxy step probably leads to peak HOM formation at somewhat higher NO conditions than non-cyclic VOCs, and stating explicitly that multiple alkoxy-peroxy steps are not necessary for HOM formation.

325 In line 400 in the revised manuscript:

"Though further alkoxy-peroxy steps are not necessary for HOM formation from limonene, subsequent autoxidation steps will compete against bimolecular reactions with NO, suggesting that more alkoxy-peroxy steps may occur even when the unimolecular lifetime of these later non-cyclic $RO_2$ radicals is typically shorter than that of the early-stage cyclic $RO_2$."

330 In line 410 in the revised manuscript:

"The need of a ring-breaking alkoxy-peroxy step in the proposed HOM formation mechanism does suggest that the highest yields of limonene HOM formation may occur at slightly higher NO concentrations than for non-cyclic VOCs who autoxidize even without alkoxy steps."

335 *How good is the assumption to use "the same kuni, kRO2+RO2, kRO2+HO2, and α for a given C10H15Ox• and C10H17Ox• family."? Easy examples where this does not hold are: H-shift across the ring vs H-shift in a linear chain after ring breaking. RO2 + RO2 for a primary RO2 vs oxygenated RO2. How big of an uncertainties these assumption generate?*

**Response:**

340 We agree the assumption can lead to uncertainties. Both the $C_{10}H_{15}O_x\bullet$ and $C_{10}H_{17}O_x\bullet$ families include primary or oxygenated $RO_2$. The k of $RO_2$ reaction cannot be determined directly by experiment at present.

As mentioned in the manuscript, $C_{10}H_{16}O_x$ can be divided into carbonyls (RC=O) and alcohols (ROH) or hydroperoxides (ROOH) from $C_{10}H_{17}O_x\bullet$ reactions. When $k_{uni}$, $k_{RO2+RO2}$, $k_{RO2+HO2}$, and α were

345 assigned of (0.01-1)*10$^{-12}$ cm$^3$ molecule$^{-1}$ s$^{-1}$ (Crounse et al., 2013), (0.001-1)*10$^{-10}$ cm$^3$ molecule$^{-1}$ s$^{-1}$ (Berndt et al., 2018), and (0.5-2)*10$^{-11}$ cm$^3$ molecule$^{-1}$ s$^{-1}$ (Ziemann and Atkinson, 2012), respectively, one can get the yield of carbonyl according to Eq. 6 and Eq. 7, which ranged from 90%-96% at low NO and 97%-100% at high NO. This indicated that the yields of carbonyl are not sensitive to these assumption k and α.

350 We revised the equations and expanded our discussion in line 206 in the revised main text:

"We did a sensitivity analysis to test the influence of varying the $k_{uni}$, $k_{RO2+RO2}$, $k_{RO2+HO2}$, and α using the ranges of these parameters reported in the literature on the fraction of carbonyl in $C_{10}H_{16}O_x$ and on the importance of H-abstraction channel in HOM formation. When $k_{uni}$, $k_{RO2+RO2}$, $k_{RO2+HO2}$, and α were varied in the range of (0.01-1)*10$^{-12}$ cm$^3$ molecule$^{-1}$ s$^{-1}$, (0.001-1)*10$^{-10}$ cm$^3$ molecule$^{-1}$ s$^{-1}$, (0.5-2)*10$^{-11}$

355 cm$^3$ molecule$^{-1}$ s$^{-1}$ based on the values in the literature (Crounse et al., 2013; Berndt et al., 2018; Ziemann and Atkinson, 2012), and 0.5, respectively, one can get the yield of carbonyl according to Eq. 6 and Eq. 7, which ranged from 90%-96% at low NO and 97%-100% at high NO. This indicated that the yields of carbonyl are not sensitive to these assumption of k and α."

We have further estimated the uncertainty of fraction of $C_{10}H_{15}O_x\bullet$-related products in C10-HOM resulted from the allocation of carbonyls and alcohols in $C_{10}H_{16}O_x$ in Eq.2-8. The contributions of $C_{10}H_{15}O_x\bullet$-related products range from 39.5% to 41.4% at low NO and 42.2% to 42.6% at high NO, respectively. We found that fraction of $C_{10}H_{15}O_x\bullet$ related products in $C_{10}$-HOM was not much affected by how carbonyls and alcohols in $C_{10}H_{16}O_x$ is allocated. We expanded our discussion in ==line 507== in the revised main text:

"We have further estimated the uncertainty of fraction of $C_{10}H_{15}O_x\bullet$-related products in C10-HOM resulted from the allocation of carbonyls and alcohols in $C_{10}H_{16}O_x$ in Eq.2-8. The contributions of $C_{10}H_{15}O_x\bullet$-related products range from 39.5% to 41.4% at low NO and 42.2% to 42.6% at high NO, respectively. We found that fraction of $C_{10}H_{15}O_x\bullet$ related products in $C_{10}$-HOM was not much affected by how carbonyls and alcohols in $C_{10}H_{16}O_x$ is allocated."

*How does the shifting detection sensitivity of CIMS affect your conclusions (e.g. Hyttinen et al. https://pubs.acs.org/doi/10.1021/acs.jpca.7b10015), as the C10H17Ox is likely detected better than the C10H15Ox products, especially at the lower end of the oxygen content?*

**Response:**

To our knowledge, there is no evidence that the $C_{10}H_{17}O_x\bullet$ is detected better than the $C_{10}H_{15}O_x\bullet$ products. Currently the sensitivity of individual HOM cannot be obtained. We assume the similar detection sensitivity to $C_{10}H_{15}O_x\bullet$ related HOM and $C_{10}H_{17}O_x\bullet$ related HOM. The basis of this premise is that $C_{10}H_{15}O_x\bullet$ and $C_{10}H_{17}O_x\bullet$ related HOM with six and more oxygen atoms from autooxidation (Bianchi et al., 2019; Mentel et al., 2015) have two or more hydrogen bond donors (-OOH or -OH), which allows HOM to form strong clusters with $NO_3^-$. These clusters need be competitive to the very stable $(HNO_3)NO_3^-$ to be detected, and has the similar rates near the collision limit to that of $H_2SO_4$, resulting the same high sensitivity (Hyttinen et al., 2015). Moreover, counterparts of $C_{10}H_{15}O_x\bullet$ related HOM and $C_{10}H_{17}O_x\bullet$ related HOM with the same oxygen number, either radicals, or termination products (carbonyls or organic nitrates), only differ in the chemical structures by one C=C bond or an endocyclic peroxide ring. Therefore, the multiple H bonding based sensitivities of $C_{10}H_{15}O_x\bullet$ related- and $C_{10}H_{17}O_x\bullet$ related families are supposed to be similar (Hyttinen et al., 2017). Besides, no dependence of sensitivity on the functional groups of HOM within a maximum uncertainty of a factor of 2 was detected by using the same $NO_3^-$-CIMS with the same setting in our previous study (Pullinen et al., 2020).

Additionally, we observed that $C_{10}H_{15}O_X\bullet$ related products are comparable to $C_{10}H_{17}O_x\bullet$ related products at all oxygen numbers (Fig. S6). This indicates that the significance of $C_{10}H_{15}O_x\bullet$ related products is not affected by the detection sensitivity, which would mostly affect the sensitivity of less oxygenated compounds as the reviewer pointed out. Therefore, we conclude that the sensitivity of $NO_3^-$-CIMS to $C_{10}H_{15}O_x\bullet$ and $C_{10}H_{17}O_x\bullet$ related HOM is the same and near the collision limit.

If the sensitivity of $C_{10}H_{17}O_x\bullet$ related products were indeed higher than $C_{10}H_{15}O_X\bullet$ related products, the $C_{10}H_{15}O_X\bullet$ related products would be underestimated and they would be even more important than the current estimate. This will not change our conclusion that the $C_{10}H_{15}O_x\bullet$ related products contribute significantly to HOM formation.

In the revised manuscript, we discussed extend the discussion in ==line 486== in the revised main text:

"Currently, an absolute calibration using HOM standards is not possible mainly due to the difficulty to synthesize pure HOM and unclear chemical structures of many HOM. However, we think that it is reasonable to expect a generally similar sensitivity for HOM in this study for the following reasons. First, Hyttinen et al. (2017) found that the increase in binding energy with $NO_3^-$

for molecules with an additional hydroxyperoxy group to two hydrogen bond donor functional groups is small for HOM formed in cyclohexene ozonolysis. As HOM in this study generally contain more than two hydrogen bond donor functional groups, their sensitivity is expected to be similar. We used a unified $H_2SO_4$-based calibration coefficient for HOM, which is commonly used to calibrate $NO_3^-$-CIMS (Kirkby et al., 2016; Jokinen et al., 2015; Rissanen et al., 2014; Ehn et al., 2014). Second, although underestimation of certain HOM $RO_2$ formed from α-pinene+OH reaction has been reported (Berndt et al., 2016), such underestimation was mainly attributed to the steric hinderance in forming HOM-nitrate cluster for HOM with bicyclic structures ($C_{10}H_{17}O_7\bullet$) and thus not common for all HOM. In our study, we found the significance of $C_{10}H_{15}O_x\bullet$-related product at all oxygen contents, particularly for closed-shell products with number of oxygen atom great than 8, indicative of more H-donating functional groups (Fig. S6 in revised Supplement). This indicates that the significance of $C_{10}H_{15}O_x\bullet$ related products is not affected by the detection sensitivity, which would mostly affect the sensitivity of less oxygenated compounds. And the presence of NO particularly at high NO leads to ring-opening reactions as shown in Scheme 1. Therefore, the HOM products from OH addition in this study are likely to form stable clusters with nitrate and thus have similar sensitivity with HOM formed via H-abstraction in nitrate CIMS. Third, our previous study showed that using an unified sensitivity of $H_2SO_4$ only leads to a maximum uncertainty of a factor of two by comparing the condensation HOM and corresponding increase of aerosol mass (Pullinen et al., 2020). If for some currently unknown reason $C_{10}H_{17}O_x\bullet$-related products had higher sensitivity than $C_{10}H_{15}O_x\bullet$-related products, this would lead to under-estimate of the significance of OH H-abstraction pathway. This will not change our conclusion that the $C_{10}H_{15}O_x\bullet$ related products contribute significantly to HOM formation."

[Figure]

**Figure S6.** Relative abundances of individual products at low NO (0.06 - 0.1 ppb, panel a-c) and high NO (~17 ppb, panel d-f) at the first 15 min with the panels (a, d) showing $C_{10}H_{15}O_x\cdot$ (peroxy radicals, x=6-15) and $C_{10}H_{17}O_x\cdot$ (peroxy radicals, x=6-15) in black, the panels (b, e) showing $C_{10}H_{14}O_x$ (carbonyls, x=7-16) and $C_{10}H_{16}O_x$ (carbonyls, x=6, 8-15) in red, and the panels (c, f) showing $C_{10}H_{15}NO_x$ (organic nitrates, x=8-16) and $C_{10}H_{17}NO_x$ (organic nitrates, x=6-15) in blue. $C_{10}H_{15}O_x\cdot$ and their related products are in solid bars and $C_{10}H_{17}O_x\cdot$ and their related products are in transparent bars. The individual products are normalized to the signals the most abundant individual product respectively ($C_{10}H_{16}O_8$ at low NO and $C_{10}H_{17}NO_9$ at high NO).

*It seems unlikely that the SAR will reproduce the rates of the highly oxygenated molecule Hshift rates. This is due to intramolecular interactions (mainly H-bonding) that influences the thermochemistry; a favorable interaction on the reactant side will increase the barrier to reaction, and can significantly decrease the H-shift rate. Looking at the schemes of the paper with the given SAR rates can leave a very wrong picture for the reader. This influence is hinted in the manuscript but, still the uncertain numbers are presented in the schemes. Is this reasonable?*

**Response:**

It is clearly stated throughout the manuscript that these numbers, and the mechanism itself, is not a quantitative model for various reasons: the uncertainties on the numbers, the selection of only one starting

RO$_2$, the selection of only one pathway from each intermediate etc. Still, the numbers given by the SAR reflect the current knowledge available for such reactions, and these best-available estimates suggest that the rates are of the correct order of magnitude to sustain autoxidation without the need to make adjustments or artificially increasing rates. We feel it is reasonable, even required, to provide readers with these current-knowledge estimates to show that the proposed reaction steps are at least viable, as far as anyone can tell at the moment.

In this class of reaction, it is often the case that both reactant, TS, and products have similar number and strengths of H-bonds, especially when there are multiple combinations of H-bonds possible; the most favorable H-bonding combination need not be the same in reactant and TS. Also, H-migration can occur even if the peroxy moiety is H-bonded. Furthermore, H-bonded structures, while energetically more favorable, are entropically less favorable such that both more and less H-bonded conformers contribute significantly to the population of both reactant and TS. In some cases, H-bonding can indeed hamper the H-migrations, but in other cases it can promote H-migration (e.g., by increasing the population contribution of conformers with the radical site near an abstractable H-atom, or favoring a cyclic TS that is best for H-migration). Overall, the complexity of H-bonding in multi-functionalized species definitely leads to larger uncertainty margins on the rate estimates, but does not imply that the SAR predictions are fundamentally biased towards too-high values and would mislead the reader.

*The secondary OH reaction is generally inherently less likely in lab systems, as I would expect to be here as well (i.e., how much would you need to accumulate the product before it'll find the second OH at a rate sufficient for measurable product formation?). Yet, with the missing documentation about the used concentration ranges it is not possible to even roughly estimate this fraction.*

**Response:**

We agree that in our study, particularly in the first 15 min, the secondary OH reaction is unlikely. We have addressed this comment in our response to the comment #1 (line 42-78 of this document). We added the concentrations in the main text (see Table S1 in the Supplement and line 143 in the main text).

*Line 323: "However, this extra bimolecular OH reaction takes time, and can delay formation of HOM. An analysis of the secondary chemistry is outside the scope of this work, …" As mentioned above, not all pathways lead to HOM, and the whole paper is an analysis of secondary chemistry if you are studying OH abstraction vs OH addition through a complex mechanism. Please sharpen your words and thinking.*

**Response:**

As we explained above (line 42-78 of this document), now we focus on the early stages of the experiments (first 15 min), and thus secondary chemistry is not important in this study. This can be quantified by the comparison of the chemistry of the limonene and limonaldehyde referring to the response to comment #1.

In the revised manuscript, we expanded the discussion in line 379 in the main text:

"In principle, $C_{10}H_{15}O_x\bullet$ peroxy radicals might also form through secondary chemistry of first-generation $C_{10}$ oxidation products of the limonene+OH reaction. The limonaldehyde ($C_{10}H_{16}O_2$) is the most abundant (99%) first-generation $C_{10}$ product reported in limonene+OH reaction (Hakola et al., 1994; Larsen et al., 2001), which can form $C_{10}H_{15}O_4\bullet$ and the $C_{10}H_{15}O_x\bullet$ family by further autoxidation through H-abstraction and subsequent $O_2$ addition. Therefore, we take limonaldehyde into account as the most competitive candidate. For the early stages of our experiments (first 15 min), however, we find that secondary chemistry is not important (Section S2 and Fig. S9 in Supplement)."

We added the comparison of limonene+OH and limonaldehyde+OH in the Section S2 and Figure S9 in the revised supplement.

490 *Line 328: Here you imply that the low NO is actually the highest of your reported range (0.2 ppb) and quote a value of 29% for RONO2 which is not in line with the previously given 33%. At these conditions this seems like a very high value. Have you considered, for example, that your CIMS could be more sensitive to nitrates than to HOM devoid of -ONO2? There are some hints in the recent literature about this.

495 **Response:**
We have corrected the error of NO concentration and the inconsistent ON fraction. ONs account for ~31% at low NO in this study. According to the $RO_2$ loss rate (Figure S2), $RO_2$+NO was the dominant $RO_2$ loss pathway at both low and high NO, although there is a very small amount (<1%) $RO_2$ loss through $RO_2$+$HO_2$ pathway at low NO. Therefore, it is not surprising that the fraction of ON in the products can
500 reach 31% at low NO. To our knowledge, there's no direct measurement evidence yet that the CIMS could be more sensitive to nitrates than to HOM devoid of -$ONO_2$. Even if the sensitivity were different, as we compared both the $C_{10}H_{15}O_x$ related HOM-ON and non-nitrate HOM with their counterparts related to $C_{10}H_{15}O_x$, it would not affect our conclusion regarding the importance of OH abstraction.

In the revised manuscript, we correct the ON fraction in line 616:
505 "Even at ~0.2 ppb NO, ONs account for a large part (31%) of HOM ONs."

*Finally, it is very difficult for me to see why higher NO can lead to higher abstraction vs addition rates. These issues seem uncoupled (i.e., no amount NO can increase an OH abstraction rate) and rather point into being misunderstood RO2 + NO (or indeed RO + NO) chemistry connected to the long processing
510 time-scales.
**Response:**
We agree with the reviewer that NO does not directly affect the H abstraction rate and we did not state that higher NO can lead to higher abstraction vs addition rates in our manuscript. What NO affects in this study is the HOM formation via H abstraction versus OH addition, which involves $RO_2$ auto-oxidation.
515 The reaction of NO with $RO_2$ promotes alkoxy radical formation which can affect the autoxidation pathway in both H abstraction channel and OH addition channel. In order to avoid potential misunderstanding, we have revised the sentence regarding the NO influence in line 533 in the revised manuscript as follows:
"…we speculate that an NO dependence of ratio of HOM formed via H-abstraction versus OH
520 addition might become apparent even in the limonene system at strongly reduced NO levels."

*Minor comments:*
*Consider changing the "dimer" into "accretion product".*
**Response:**
525 Dimers can be the accretion products between two monomers. Both words represent the product in our study. Considering contrasting with "monomer", we prefer to use "dimer".

*Line 66: "Biomolecular" is an error. (Other places as well).*
**Response:**
530 Accepted. We have corrected this typo in the main text.

*Line 150: What "organic" are you referring to? The identity matters a lot, as has been shown in this chamber setup previously.*

**Response:**

To classify, we added a description in line 168 in the revised main text:

"…organic vapor (such as $C_{10}H_{15}NO_{9-12}$ (nitrated compounds) and $C_{10}H_{14}O_{8-11}$ (non-nitrated compounds) in the reaction of limonene with OH in the presence of NO) concentrations in the dark (Guo et al., 2022)."

*Line 158: delta symbol is usually reserved for a changing quantity.*

**Response:**

Accepted. We have corrected this error in the main text.

*Line 210: How should one understand the statement: "Briefly, we only consider a limited oxidation network by considering all possible reaction channels, ..."*

**Response:**

We have modified this sentence in line 235 in the revised main text:

"Briefly, we only take into account a limited oxidation network by considering all possible reaction channels and select only the dominant channels, based on their rate as predicted by SARs."

*Don't mix -y and -yl endings, i.e., if you choose to use "peroxyl" then use "alkoxyl" as well.*

**Response:**

Accepted. We have unified the expression in the main text.

*Line 394: Didn't catch why you presume the disproportionation of RO2 + RO2 would favor carbonyls over alcohols.*

**Response:**

We did not mean that the disproportionation of $RO_2 + RO_2$ would favor carbonyls over alcohols. We apologize for the ambiguous sentence. In the revised manuscript, we modified this sentence in line 442 in the revised main text as follows:

"The higher abundance of carbonyl products compared to alcohol products indicates that here a large fraction of the carbonyls is not formed from $RO_2 + RO_2$ reactions (see also Fig. S3), but rather from termination reactions in HOOQOO• radicals eliminating an OH radical after an α-OOH H-atom migration, forming O=QOOH."

*Plural of formula is formulae.*

**Response:**

We looked up the dictionary and found that both formulas and formulae can be used as the plural form of formula. Therefore, we kept the usage of "formulas".

[revised manuscript text omitted]

---

## Author Comment (AC2)

We thank the reviewer for the helpful comments on our manuscript. The comments are greatly appreciated. We have addressed all the comments and believe that the revisions based on the comments help improve the quality of our manuscript. Below please find our responses to the comments one by one and the corresponding revisions made to the manuscript. The original comments are in italics. The revised

5    parts of the manuscript are in blue.

*Overall comment:*

*This work by Luo et al. examined HOM formation from limonene + OH and highlighted the importance of the H-abstraction pathways, in addition to the well-known OH-addition pathways. The main*

10    *conclusion is well supported by the observations of C10H15Ox-RO2 as well as their termination products. These results are also in line with the authors' recent publication on H-abstraction derived HOM for alpha-pinene + OH chemistry (Shen et al., 2022 Sci. Adv.). Therefore, I believe that the underscored H-abstraction pathways for monoterpene + OH chemistry are important and should be considered in future chemical mechanisms. However, there are a number of major issues that need to be*

15    *addressed before consideration for publication, as discussed below.*

*Detailed comments:*

*1. Estimation of HOM yields. This work used NO3-CIMS to estimate HOM yields and relative contributions of the H-abstraction vs. OH-addition pathways, assuming unified sensitivity for all the*

20    *HOMs products. However, this approach may have large uncertainties. For example, Berndt et al. (Nature Comm. 2016, 7, 13677) demonstrated that NO3-CIMS could underestimate HOM formation from OH-addition pathways for alpha-pinene and beta-pinene. This might also be the case for limonene. If this is true, then the contribution of OH-addition to limonene HOMs as well as the total HOM yields could be underestimated. Better constraints on these aspects are needed.*

25    **Response:**

We thank the reviewer for the supportive remarks. Regarding the sensitivity, admittedly, using unified sensitivity may lead to uncertainties. Currently, an absolute calibration using HOM standards is not possible mainly due to the difficulty to synthesize pure HOM and unclear chemical structures of many HOM.

30        However, we think that it is reasonable to expect a generally similar sensitivity for HOM in this study for the following reasons. First, Hyttinen et al. (2017) found that the increase in binding energy with $NO_3^-$ for molecules with an additional hydroxyperoxy group to two hydrogen bond donor functional groups is small for HOM formed in cyclohexene ozonolysis. As HOM in this study generally contain more than two hydrogen bond donor functional groups, their sensitivity is expected to be similar. We

35    used a unified $H_2SO_4$-based calibration coefficient for HOM, which is commonly used to calibrate $NO_3^-$-CIMS (Kirkby et al., 2016; Jokinen et al., 2015; Rissanen et al., 2014; Ehn et al., 2014). Second, the underestimation of total HOM $RO_2$ concentrations from the OH radical reaction using nitrate ionization while not for HOM formed by ozonolysis reported by Berndt et al. (2016) is mostly contributed by a single HOM-$RO_2$ $C_{10}H_{17}O_7\bullet$. The reason for the difference in sensitivity between HOM formed in two

40    oxidation scheme was attributed mainly to the steric hindrance in forming HOM nitrate cluster for HOM with bicyclic structure ($C_{10}H_{17}O_7\bullet$). In our study, we found the significance of $C_{10}H_{15}O_x\bullet$-related product at all oxygen content and most of them were contributed by closed-shell products with number of oxygen atom great than 8, indicative of more H-donating functional groups (Fig. S6 in revised Supplement). And the presence of NO particularly at high NO leads to ring-opening reactions as shown in Scheme 1.

45    Therefore, the HOM products from OH addition in this study are likely to form stable clusters with nitrate and thus have similar sensitivity with HOM formed via H-abstraction in nitrate CIMS. Third, our previous study showed that using an unified sensitivity of $H_2SO_4$ only leads to an uncertainty of a factor of two by comparing the condensation HOM and corresponding increase of aerosol mass (Pullinen et al., 2020).

50    Nevertheless, in the revised manuscript, we added discussion regarding the influence of the sensitivity in line 486 in the revised main text:

"Currently, an absolute calibration using HOM standards is not possible mainly due to the difficulty to synthesize pure HOM and unclear chemical structures of many HOM. However, we think that it is reasonable to expect a generally similar sensitivity for HOM in this study for the
55    following reasons. First, Hyttinen et al. (2017) found that the increase in binding energy with $NO_3^-$ for molecules with an additional hydroxyperoxy group to two hydrogen bond donor functional groups is small for HOM formed in cyclohexene ozonolysis. As HOM in this study generally contain more than two hydrogen bond donor functional groups, their sensitivity is expected to be similar. We used a unified $H_2SO_4$-based calibration coefficient for HOM, which is commonly used to
60    calibrate $NO_3^-$-CIMS (Kirkby et al., 2016; Jokinen et al., 2015; Rissanen et al., 2014; Ehn et al., 2014). Second, although underestimation of certain HOM $RO_2$ formed from α-pinene+OH reaction has been reported (Berndt et al., 2016), such underestimation was mainly attributed to the steric hinderance in forming HOM-nitrate cluster for HOM with bicyclic structures ($C_{10}H_{17}O_7\bullet$) and thus not common for all HOM. In our study, we found the significance of $C_{10}H_{15}O_x\bullet$-related product at
65    all oxygen contents, particularly for closed-shell products with number of oxygen atom great than 8, indicative of more H-donating functional groups (Fig. S6 in revised Supplement). This indicates that the significance of $C_{10}H_{15}O_x\bullet$ related products is not affected by the detection sensitivity, which would mostly affect the sensitivity of less oxygenated compounds. And the presence of NO particularly at high NO leads to ring-opening reactions as shown in Scheme 1. Therefore, the HOM
70    products from OH addition in this study are likely to form stable clusters with nitrate and thus have similar sensitivity with HOM formed via H-abstraction in nitrate CIMS. Third, our previous study showed that using an unified sensitivity of $H_2SO_4$ only leads to a maximum uncertainty of a factor of two by comparing the condensation HOM and corresponding increase of aerosol mass (Pullinen et al., 2020). If for some currently unknown reason $C_{10}H_{17}O_x\bullet$-related products had higher
75    sensitivity than $C_{10}H_{15}O_x\bullet$-related products, this would lead to under-estimate of the significance of OH H-abstraction pathway. This will not change our conclusion that the $C_{10}H_{15}O_x\bullet$ related products contribute significantly to HOM formation."

*2. RO2 chemistry. In Page 2, the RO2 reactions are listed. However, there are a few missing pathways.*
80    *First, the RO2 + HO2 reaction may only partially form ROOH and could also produce RO (Kurten et al., JPCL, 2017, 8, 2826). The branching ratios of the two pathways likely depend on the RO2 structure. In addition, RO unimolecular isomerization could produce RO2 in the same CHO family. For example, C10H15Ox RO could form C10H15O(x+2) RO2. Therefore, this pathway needs to be included in the data analysis. Lastly, the RO2 + RO2 reaction described in (R2) is only correct for primary and*
85    *secondary RO2. For tertiary RO2s, carbonyls cannot be formed. Without considering these above-mentioned pathways, the estimation of contributions from C10H15Ox vs. C10H17Ox to C10H16Ox (i.e., Eq. 2 – Eq. 8) is wrong.*
**Response:**

We agree with the comment. In the revised manuscript, we have added the reaction of $RO_2 + HO_2$ forming RO, the unimolecular reaction of RO forming carbonyl and clarified the validity of $RO_2 + RO_2$ (R2). We have revise pathways in page 2 in the main text and Eq. 3 in line 185 in the revised main text:

"R1a, which forms ROOH with a yield β , where β is close to 1 for most $RO_2$ (Jenkin et al., 2019).

$$\frac{d[ROOH]}{dt} = k_{RO2+HO2}[RO_2][HO_2]\beta \qquad \text{(Eq.3)}"$$

Also, we added more detail description in line 75 in the revised main text:

"Note that $RO_2$ reaction in R2 is considered for primary and secondary $RO_2$. For tertiary $RO_2$, carbonyls cannot be formed. In addition, the unimolecular isomerization of RO (from R1b and R3-4) could produce $RO_2$ in the same $RO_2$ family."

We moved a description from Scheme 2 to line 385 in the revised main text, as shown below:

"Direct autoxidation of the nascent $RO_2$ is slow, $k=\sim10^{-2}$ $s^{-1}$, and formation of an alkoxy radical is to be expected immediately or after very few autoxidation steps, especially in high NO conditions. Once the ring structure is broken, fast autoxidation steps are accessible. All $RO_2$ intermediates have competing reactions (not shown) under current conditions with $HO_2$ (forming hydroperoxides) and NO (forming alkoxy radicals and nitrates). Alkoxy radicals formed thus can fragment, or continue autoxidation after ring breaking or fast migration of an hydroperoxide H-atom, forming a wider variety of HOM."

We have further estimated the uncertainty of fraction of $C_{10}H_{15}O_x\bullet$-related products in C10-HOM resulted from the allocation of carbonyls and alcohols in $C_{10}H_{16}O_x$ in Eq. 2-8. The contributions of $C_{10}H_{15}O_x\bullet$-related products range from 39.5% to 41.4% at low NO and 42.2% to 42.6% at high NO, respectively. We found that fraction of $C_{10}H_{15}O_x\bullet$ related products in $C_{10}$-HOM was not much affected by how carbonyls and alcohols in $C_{10}H_{16}O_x$ is allocated. We expanded our discussion in line 507 in the revised main text:

"We have further estimated the uncertainty of fraction of $C_{10}H_{15}O_x\bullet$-related products in C10-HOM resulted from the allocation of carbonyls and alcohols in $C_{10}H_{16}O_x$ in Eq.2-8. The contributions of $C_{10}H_{15}O_x\bullet$-related products range from 39.5% to 41.4% at low NO and 42.2% to 42.6% at high NO, respectively. We found that fraction of $C_{10}H_{15}O_x\bullet$ related products in $C_{10}$-HOM was not much affected by how carbonyls and alcohols in $C_{10}H_{16}O_x$ is allocated."

*3. Products in the low-NO conditions. A range of NO concentration was reported (0.1 ppt – 0.2 ppb) for the low-NO conditions. So, I assume there were multiple experiments performed under low-NO conditions with varied NO concentrations from as low as 0.1 ppt up to 0.2 ppb. Then, I would expect the product distributions under varied NO concentrations to be reported. However, in Figures 2 and 3, only one set of results were shown for low-NO. What is the NO concentration for the low-NO conditions shown? It is even more confusing that the fractions of C6-C9 fragmentation products (Figure 2) and organic nitrates (Figure 3) are not significantly lower for the low-NO experiments in comparison to the high-NO experiments. Even at 0.2 ppb for low-NO, it is ~ 100 times lower than the high-NO condition. But the fragmentation products are only different by 7%, and organic nitrates are 34% vs. 73%. More thorough analysis is needed to verify the difference.*

**Response:**

We thank the reviewer for the helpful comment.

"0.1 ppt" is a typo, which was corrected in line 139 in the revised main text:

"0.06 - 0.1 ppb"

$RO_2$+NO is the dominant pathway of $RO_2$ loss at both low and high NO (Figure S3).

To avoid the effects of second generation chemistry, we focus on the first 15min in our revised manuscript. After we updated the data, the ratio of fraction of fragmentation product at low NO to that at high NO is similar to the ratio of the fraction of organic nitrates. During the first 15 min of the experiments, at low and high NO the fraction of fragmentation products (C6-9 HOM) in total HOM are 24.1% and 45.5%, respectively. This is consistent with the fraction of C10 organic nitrates in C10 monomers of (28.1% and 55.4% at low and high NO, respectively), which indicates that high NO concentration is conducive to the generation of organic nitrates and fragment products.

We updated the ratios and added more discussion in line 98 in the revised supplement:

"Within the first 15 min of the experiments, at low and high NO the ratios of $\frac{C_{6-9}\ monomers}{total\ HOM\ products}$

are 0.24 and 0.46, respectively. This is consistent with the ratios of $\frac{C_{10}H_{15}NO_x + C_{10}H_{17}NO_x}{C_{10}\ monomers}$ (0.28 and 0.55

at low and high NO, respectively), which indicates that high NO concentration is conducive to the generation of organic nitrates and fragment products."

*4. Product distributions. In Figure 3, what is there a strong Nox dependence for the C10H15Ox/C10H17Ox ratio?*

**Response:**

We guess that the reviewer meant "why" by "what". The $NO_x$ dependence for $C_{10}H_{15}O_x\bullet/C_{10}H_{17}O_x\bullet$ may be attributed to the differences in reactivity of them. One explanation to the $NO_x$ dependence is that the autooxidation of $C_{10}H_{15}O_x\bullet$ $RO_2$ radicals may be faster than that of $C_{10}H_{17}O_x\bullet$ $RO_2$ radicals, which lead to the lower concentration of $C_{10}H_{15}O_x\bullet$ observed in this study and higher sensitivity to NO concentrations. Assuming a steady state for highly oxygenated $C_{10}H_{15}O_x\bullet$ $RO_2$ and $C_{10}H_{17}O_x\bullet$ $RO_2$. Based the production rate and loss rate, one can get the following equation for $C_{10}H_{15}O_x\bullet$.

$$k[LIM][OH]Y_1\alpha = k_{uni\_1}[C_{10}H_{15}O_x\bullet] + k_1[C_{10}H_{15}O_x\bullet][NO] \qquad \text{Eq. R1}$$

where k is the reaction rate constant for the reaction of limonene+OH, $k_{uni\_1}$ is the average unimolecular reaction rate constant of $[C_{10}H_{15}O_x\bullet]$ as a whole, $k_1$ is the rate constant for the reaction of $C_{10}H_{15}O_x\bullet$+NO. $\alpha$ is the branching ratio of OH abstraction and $Y_1$ is fraction of primary $RO_2$ undergoing auto-oxidation to form highly oxygenated $C_{10}H_{15}O_x\bullet$.

From Eq. R1, one can get

$$[C_{10}H_{15}O_x\bullet] = k[LIM][OH]Y_1\alpha/(k_{uni\_1} + k_1[NO]) \qquad \text{Eq. R2}$$

Similarly, one can get the following equation for $C_{10}H_{17}O_x\bullet$.

$$k[LIM][OH]Y_2(1-\alpha) = k_{uni\_2}[C_{10}H_{17}O_x\bullet] + k_2[C_{10}H_{17}O_x\bullet][NO] \qquad \text{Eq. R3}$$

where $k_{uni\_2}$ is the average unimolecular reaction rate constant of $[C_{10}H_{17}O_x\bullet]$ as a whole, $k_2$ is the rate constant for the reaction of $C_{10}H_{17}O_x\bullet$+NO. $\alpha$ is the branching ratio of OH abstraction and $Y_2$ is fraction of primary $RO_2$ undergoing auto-oxidation to form highly oxygenated $C_{10}H_{17}O_x\bullet$.

$$[C_{10}H_{17}O_x\bullet] = k[LIM][OH]\ Y_2(1-\alpha)/(k_{uni\_2} + k_2[NO]) \qquad \text{Eq. R4}$$

k1 and k2 can be considered equal (Ziemann and Atkinson, 2012).

From Eq. R2 and R4, one can get the ratio

$$\frac{[C_{10}H_{15}O_x\ \bullet]}{[C_{10}H_{17}O_x\ \bullet]} = \frac{\alpha Y_1}{(1-\alpha)Y_2}(1 + \frac{k_{uni\_2} - k_{uni\_1}}{k_1[NO] + k_{uni\_1}})$$

When $k_{uni\_1} > k_{uni\_2}$, the ratio of $[C_{10}H_{15}O_x\bullet]$ to $[C_{10}H_{17}O_x\bullet]$ increases with increasing NO concentration. In the revised manuscript, we have added the following text in line 305 in the revised main text:

"The $NO_x$ dependence for $C_{10}H_{15}O_x\bullet/C_{10}H_{17}O_x\bullet$ may be attributed to the differences in their

reactivity. One explanation to the $NO_x$ dependence is that the autooxidation of $C_{10}H_{15}O_x\bullet$ $RO_2$ radicals may be faster than that of $C_{10}H_{17}O_x\bullet$ $RO_2$ radicals, which lead to the lower concentration of $C_{10}H_{15}O_x\bullet$ observed in this study and higher sensitivity to NO concentrations."

*Why are the ratios of C10H18Ox/C10H17Ox fairly constant (28% vs. 19%) between the two NO conditions, but C10H16Ox(R-OH/OOH)/C10H15Ox ratios are very different (51% vs. 0%)?*

**Response:**

It is an error, and $C_{10}H_{16}O_x$(R-OH/OOH) accounts for 0.2% of C10 HOMs at high NO in the first 15 min. We correct it and update Figure 3, and also added a description in line 312 in the revised main text:

"When we focus on the first 15 min of experiments to avoid the influence of secondary chemistry, ratios of $C_{10}H_{16}O_x$(R-OH/OOH)/$C_{10}H_{15}O_x\bullet$ can be derived of 0.47 and 0.02 at low and high NO, respectively. The ratio of $C_{10}H_{18}O_x$/$C_{10}H_{17}O_x\bullet$ at low NO (0.38) is higher than at high NO (0.1)."

*The relationships between different families under different NO conditions need to be discussed in-depth.*

**Response:**

In our revised manuscript, we focus on the first 15 min of experiments to avoid the influence of secondary chemistry. Then one can get the ratios of $C_{10}H_{16}O_x$(R-OH/OOH)/$C_{10}H_{15}O_x\bullet$ are 0.47 and 0.02 at low and high NO, respectively. The ratio of $C_{10}H_{18}O_x$/$C_{10}H_{17}O_x\bullet$ at low NO (0.38) is higher than that at high NO (0.1). The decrease of $C_{10}H_{16}O_x$(R-OH/OOH)/$C_{10}H_{15}O_x\bullet$ at high NO compared to low NO was more evident than the decrease of $C_{10}H_{18}O_x$/$C_{10}H_{17}O_x\bullet$. Theoretically, they should be similar. Such a difference may be attributed to the shift in $C_{10}H_{15}O_x\bullet$ distribution with different number of O and isomers at high NO compared to low NO. The shift of in $C_{10}H_{15}O_x\bullet$ distribution with different number of O is evident in Fig. S6. At high NO there might be more $C_{10}H_{15}O_x\bullet$ that react slower with $HO_2$ or have a lower branching ratio forming ROOH in $RO_2$+$HO_2$, which depends on detailed $RO_2$ structure as mentioned by the reviewer or have a lower yield forming ROH in $RO_2$+$RO_2$.

In the revised manuscript, we have discussed this issue as follows.

We expanded our discussion in line 305 in the revised main text:

"The $NO_x$ dependence for $C_{10}H_{15}O_x\bullet$/$C_{10}H_{17}O_x\bullet$ may be attributed to the differences in their reactivity. One explanation to the $NO_x$ dependence is that the autooxidation of $C_{10}H_{15}O_x\bullet$ $RO_2$ radicals may be faster than that of $C_{10}H_{17}O_x\bullet$ $RO_2$ radicals, which lead to the lower concentration of $C_{10}H_{15}O_x\bullet$ observed in this study and higher sensitivity to NO concentrations. Based on the measured concentrations of $RO_2\bullet$, $HO_2\bullet$, and NO, an average bimolecular $RO_2\bullet$ loss rate of ~0.02 $s^{-1}$ (low NO) and ~3.5 $s^{-1}$ (high NO) was estimated in our previous study (Zhao et al., 2018), which is predominately due to the reaction with NO. From this, we infer that the average reactive rate of $C_{10}H_{17}O_x\bullet$ at high NO is higher than that at low NO, which finally result in the increases of $C_{10}H_{17}O_x\bullet$ consumption. This inference is supported by the higher relative contribution of $C_{10}H_{17}NO_x$ at high NO (36.3%) than at low NO (16.1%) in Fig. 2. When we focus on the first 15 min of experiments to avoid the influence of secondary chemistry, ratios of $C_{10}H_{16}O_x$(R-OH/OOH)/$C_{10}H_{15}O_x\bullet$ can be derived of 0.47 and 0.02 at low and high NO, respectively. The ratio of $C_{10}H_{18}O_x$/$C_{10}H_{17}O_x\bullet$ at low NO (0.38) is higher than at high NO (0.1). The decrease of $C_{10}H_{16}O_x$(R-OH/OOH)/$C_{10}H_{15}O_x\bullet$ at high NO compared to low NO was more evident than the decrease of $C_{10}H_{18}O_x$/$C_{10}H_{17}O_x\bullet$. Theoretically, though, they should be similar. The difference may be attributed to the shift in $C_{10}H_{15}O_x\bullet$ distribution with different number of O, as evident in Fig. S6, and different

isomers at high NO compared to low NO. At high NO there might thus be more $C_{10}H_{15}O_x\bullet$ that react slower with $HO_2$ or have a lower branching ratio forming ROOH in $RO_2+HO_2$, which depends on the explicit $RO_2$ structure, or have a lower yield forming ROH in $RO_2+RO_2$."

220

*5. The authors suggest that some C10H17Ox and related products may be from secondary-generation reactions. For example, the first-generation product limonaldehyde (C10H16O2) could react with OH and form C10H17Ox. In the same way, H-abstraction in limonaldehyde reacting with OH to form C10H15Ox RO2 and further products. In MCM, H-abstraction in this reaction has a branching ratio of*

225 *~28.8%. Could this reaction explain some or a major fraction of total C10H15Ox products?*

**Response:**

In the revised manuscript, we focus on the early stages of the experiments (first 15 min), when secondary chemistry is not important. This can be demonstrated by the comparison of the chemistry of the limonene and limonaldehyde. In this study, $C_{10}H_{15}O_x\bullet$ peroxy radicals can also form through $C_{10}$ first-generation

230 oxidation products in limonene+OH reaction. The limonaldehyde ($C_{10}H_{16}O_2$) is considered as the most competitive candidate, which is one of the main products in limonene+OH reaction and the most abundant first-generation $C_{10}$ product reported (Hakola et al., 1994; Larsen et al., 2001). H-abstraction in limonaldehyde+OH and subsequent $O_2$ addition could lead to $C_{10}H_{15}O_4\bullet$, which would also be an initial step to form the $C_{10}H_{15}O_x\bullet$ family by further autoxidation. While the time length was 60 min in

235 our original manuscript, in the revised manuscript we focus on the early stage of reaction (the first 15 min) in order to minimize the possible contribution from the second-generation reactions. Similarly, in this early stage, second-generation chemistry of OH with the first-generation products is negligible compared to the reaction of limonene (Fig. S9). This can be quantified using the following comparison of the respective reaction rate of OH via hydrogen abstraction with limonene and limonaldehyde, which

240 is the dominant first-generation $C_{10}$ product (99% among $C_{10}$ products). We calculated the relative reaction rate of hydrogen abstraction from limonene to that from limonaldehyde by OH radicals, as shown below:

$$\frac{R[LIM + OH]_{H\,abstraction}}{R[LIMAL + OH]_{H\,abstraction}} = \frac{k[LIM + OH] \times [LIM] \times [OH] \times BR_{LIM}[H\,abstraction]}{k[LIMAL + OH] \times [LIMAL] \times [OH] \times BR_{LIMAL}[H\,abstraction]}$$

$$= \frac{k[LIM+OH] \times [LIM] \times BR_{LIM}[H\,abstraction]}{k[LIMAL+OH] \times [LIM]_r \times Y[LIMAL] \times BR_{LIMAL}[H\,abstraction]} \qquad \text{Eq. S4}$$

245 where k[LIM+OH] and k[LIMAL+OH] are reaction rate constants (Atkinson, 1997). Here, $BR_{LIM}$[H-abstraction] and $BR_{LIMAL}$[H-abstraction] are the branching ratios for limonene + OH and limonaldehyde + OH reactions to undergo H-abstraction channel, respectively. A branching ratio of 0.34 for limonene + OH was used (Rio et al., 2010) and a ratio of 0.29 for limonaldehyde + OH was used based on MCM v3.3.1 (http://mcm.york.ac.uk/). The concentrations of limonene were directly measured while

250 concentrations of limonaldehyde were estimated according to their NO-dependent yields (Y[LIMAL] in Equation 1), with a value of 0.29 at low NO and 0.28 at high NO (Hakola et al., 1994). The uncertainties

of $\frac{R[LIM+OH]_{H\,abstraction}}{R[LIMAL+OH]_{H\,abstraction}}$ was estimated to be -41%/+141% at low NO and high NO, from the uncertainty

of limonene concentration (~15%), k[LIM+OH] ($\Delta$logk=$\pm$0.08), and Y[LIMAL] ($\pm$0.06 at low NO and high NO (Hakola et al., 1994)) using error propagation. Hydrogen abstraction from limonene is 19-1600

255 times faster than that from limonaldehyde at low NO and 29-87 times faster at high NO. Note that the concentrations of limonaldehyde were estimated from consumed limonene and yields of limonaldehyde, thus reflecting only the production. As limonaldehyde is continuously consumed by OH, its true

concentration should be lower, and thus its relative importance is even overestimated using this method.

Moreover, we ran MCM model with H abstraction of OH from limonene and obtained similar results of the relative reaction rates of OH abstraction from limonene and from limonaldehyde (Figure S9).

Overall, we thus conclude that even the dominant product limonaldehyde likely has only negligible contribution to HOM formation at early stages of the experiments. Therefore, second-generation reactions are unlikely to contribute the $C_{10}H_{15}O_x$-related HOM observed in our study at those time scales. In the revised manuscript, we expanded the discussion in line 379 in the revised main text:

"In principle, $C_{10}H_{15}O_x\bullet$ peroxy radicals might also formed through secondary chemistry of first-generation $C_{10}$ oxidation products of the limonene+OH reaction. The limonaldehyde ($C_{10}H_{16}O_2$) is the most abundant (99%) first-generation $C_{10}$ product reported in limonene+OH reaction (Hakola et al., 1994; Larsen et al., 2001), which can form $C_{10}H_{15}O_4\bullet$ and the $C_{10}H_{15}O_x\bullet$ family by further autoxidation through H-abstraction and subsequent $O_2$ addition. Therefore, we take limonaldehyde into account as the most competitive candidate. For the early stages of our experiments (first 15 min), however, we find that secondary chemistry is not important (Section S2 and Fig. S9 in Supplement)."

We added the comparison of limonene+OH and limonaldehyde+OH in the Section S2 and Figure S9 in the revised supplement:

"The importance of secondary chemistry is not important in this studythe C10H15Ox-related HOM formation. This can be demonstrated by the following comparison of the chemistry of the limonene and limonaldehyde, which is the dominant first-generation $C_{10}$ product (>99%). To quantify the relative importance of these two pathways, the relative reaction rates of hydrogen abstraction from limonene+OH to that from limonaldehyde+OH were calculated as below:

$$\frac{R[LIM + OH]_{H\ abstraction}}{R[LIMAL + OH]_{H\ abstraction}} = \frac{k[LIM + OH] \times [LIM] \times [OH] \times BR_{LIM}[H\ abstraction]}{k[LIMAL + OH] \times [LIMAL] \times [OH] \times BR_{LIMAL}[H\ abstraction]}$$

$$= \frac{k[LIM+OH] \times [LIM] \times BR_{LIM}[H\ abstraction]}{k[LIMAL+OH] \times [LIM]_r \times Y[LIMAL] \times BR_{LIMAL}[H\ abstraction]} \qquad \text{(Eq. S4)}$$

where k[LIM+OH] and k[LIMAL+OH] are reaction rate constants based on MCM v3.3.1 (Atkinson, 1997). [LIM], [LIMAL], and [OH] are the concentrations of limonene, limonaldehyde, and OH radicals, while limonene and OH radicals concentrations were measured and concentrations of limonaldehyde were estimated on the basis of their NO-dependent yields (0.29 at low NO and 0.28 at high NO) (Y[LIMAL] in Equation S4) (Hakola et al., 1994). $BR_{LIM}$[H abstraction] and $BR_{LIMAL}$[H abstraction] are the branching ratio of H-abstraction channel from limonene + OH and limonaldehyde + OH, respectively. The branching ratio is 0.34 for the reaction limonene + OH (Rio et al., 2010) and 0.29 for limonaldehyde + OH based on MCM v3.3.1 (http://mcm.york.ac.uk/). The uncertainties of the relative reaction rates were estimated to be -41%/+141% at low NO and high NO, from the uncertainty of limonene concentration (~15%), k[LIM+OH] ($\Delta\log k = \pm 0.08$), and Y[LIMAL] ($\pm 0.06$ at low NO and high NO) using error propagation. As a result, hydrogen abstraction from limonene is 19-1600 times faster than that from limonaldehyde at low NO and 29-87 times at high NO (Fig. S9). Note that the concentrations of limonaldehyde were estimated from consumed limonene, which only reflect the production and neglect consumption. Therefore, the relative importance of limonaldehyde was even overestimated using this method. Based on this evidence, the contribution of limonaldehyde to HOM formation was likely negligible at early stages of the experiments. Therefore, the second-generation reactions are unlikely to contribute the $C_{10}H_{15}O_x$-related HOM observed in our study."

[Figure]

**Figure S9.** The relative ratio of hydrogen abstraction rate of the reaction limonene+OH to that of the reaction limonaldehyde+OH within the first 15 min reaction time obtained from measured at low NO (a, c) and high NO (b, d). Panels a-b and c-d show the results obtained from measured limonene concentration and limonaldehyde yield and from MCM modeling, respectively. The dashed lines are at the value of 10 (i.e., ~10% contribution of secondary chemistry). Note that different scales of y axes between panel (a, c) and (b, d). The large change in panel (b) results from the large measurement uncertainty of low accumulated limonene consumption measured by PTR-ToF-MS in the first few minutes.

*Minor comments:*

*1. The concentration of limonene used in the experiments are not mentioned in the main text.*

**Response:**

To make it clear, we added a description to the revised main text in line 140:

"OH radicals were generated from the photolysis of HONO in both low and high NO experiments and the HONO was formed from the Teflon chamber wall via a photolytic process. The details have been described by Rohrer et al. (2005). $HO_2$ was produced from the reaction of $O_2$ with RO, which can be formed in the reaction of $RO_2$+NO in photo-oxidation during the experiments. The concentration of limonene was 7 ppb. The reaction time after the roof opened was 8 hours."

*2. Suggested references:*

*Line 72. For HOM dimers: Zhao et al. PNAS, 2018, 115, 12142.*

*Line 89. For limonene + NO3: Mayorga et al., ES&T, 2022, 56, 15337.*

**Response:**

Accepted. We have added in the line 75 and line 95 in the main text.

*3. Line 84. Should be C10H14-18O7-11.*

**Response:**

Accepted. We have corrected this typo in the main text.

*4. Line 93. Here the actual HOM yield from Jokinen et al. should be mentioned.*

**Response:**

330    We have added the actual HOM yields from Jokinen et al. (2015). In line 101 in the revised main text as follow:

"…without OH scavenger (HOM yields: 5.3% (limonene+$O_3$); 0.93% (limonene+OH)) (Jokinen et al., 2015)."

335    *5. Nox analyzer. What is the detection limit of the Nox analyzer? The manuscript says low-NO has NO concentration of 0.1 ppt – 0.2 ppb but did not clearly say that the Nox analyzer measured the NO concentration.*

**Response:**

The detection limit of the $NO_x$ analyzer is 5 and 10 pptv for NO and $NO_2$, respectively. "0.1 ppt" is a

340    typo and we corrected it in line 139 in the revised main text:

"0.06 - 0.1 ppb".

To classify, we have revised in line 127 in the revised main text:

"… the concentrations of $NO_2$, NO and $O_3$, respectively."

345    *6. Line 146-147. This sentence should move to the results section.*

**Response:**

Accepted. We move this sentence to Section 3.4 in line 550 in the revised main text.

*7. Line 163. How are epoxides formed?*

350    **Response:**

Generally, epoxides can be form from H-abstraction in β-hydroperoxy alkyl radicals and then cyclisation with elimination of OH (See a possible scheme as below, adapted from Bianchi et al. (2019)).

355    *8. In the equations, sometimes R-H=O is used for carbonyls and sometimes R=O is used. They need to be the same.*

**Response:**

Accepted. We have unified the expression in the manuscript.

360    *9. Line 394. Carbonyls may also be formed from RO + O2 if the RO is primary or secondary.*

**Response:**

We agree with the reviewer. However, for large RO (C10 in this study), the RO+$O_2$ is generally slower than unimolecular reactions including the isomerization (H-shift) and decomposition (See e.g. Vereecken and Peeters (2010, 2009)). Therefore, we think that RO + $O_2$ is not a major source of carbonyl in this

365    study. Nevertheless, we have added discussion of this point in line 442 in the revised main text as follows.

"The much higher abundance of carbonyl than alcohol is unlikely to be explained by the RO+$O_2$ forming carbonyl as for large RO ($C_{10}$ in this study), the RO+$O_2$ is generally slower than unimolecular reactions including the isomerization (H-shift, i.e., alkoxy-peroxy pathway) and decomposition."

In addition, the original sentence in line 445 was not clear and we have re-formulated the sentence

370      as follows.

"The higher abundance of carbonyl products compared to alcohol products indicates that that here a large portion of carbonyls are not formed from $RO_2 + RO_2$ reactions (see also Figure S3), but rather from termination reactions in HOOQOO• radicals eliminating an OH radical after an α-OOH H-atom migration, forming O=QOOH."

375

*10. Line 503. Molar yields? Line 505. SOA mass yields?*

**Response:**

Yes. Thanks for the comment. We added the clarification in line 583 of the revised main text:

"The molar yields…"

380      and in line 585 in the revised main text:

"…SOA mass yields…"

*11. Line 522. Significant contribution of RO2 + HO2 after 2 h at low NO. But this study only focused on within 1 h after oxidation, correct?*

385      **Response:**

Yes. In our original manuscript, we focus on the oxidation within 1 hour. However, we now focus on the first 15 min of the experiments in discussion to minimize the effects of secondary chemistry.

In the revised manuscript, we have modified this sentence in line 608 in the revised main text as follows:

390      "The major $RO_2$ loss rate in all experiments is via $RO_2$+NO."

**Reference**

Atkinson, R.: Gas-Phase Tropospheric Chemistry of Volatile Organic Compounds: 1. Alkanes and

395      Alkenes, J. Phys. Chem. Ref. Data, 26, 215-290, 10.1063/1.556012, 1997.

[revised manuscript text omitted]

---

## Author Response (AR2)

We thank the reviewer for the helpful comments on our manuscript. The comments are greatly appreciated. We have addressed all the comments and believe that the revisions based on the comments help improve the quality of our manuscript. Below please find our responses to the comments one by one and the corresponding revisions made to the manuscript. The original comments are in italics. The revised parts of the manuscript are in blue.

*Revision comment:*
*In the revision, the authors have addressed some of my comments and improved the manuscript quality overall. However, there are a few remaining issues that were insufficiently explained or discussed, as detailed below.*

*Initial comment #1:*
*The Hyttinen et al. (2017) study was to explain why less oxidized HOMs are not well detected with NO3-ionization. Thus, it is inappropriate to use this study to debate whether underestimation of HOM yields is possible. HOMs with high oxygen numbers from both alpha-pinene ozonolysis and alpha-pinene + OH oxidation have more than two hydrogen bond donor functional groups. Though, the sensitivities are dramatically different. The author further argued that the underestimation of HOMs from alpha-pinene + OH was mainly attributed to the steric hindrance in forming HOM-nitrate cluster for HOM with bicyclic structures and thus not common for all HOMs. First of all, is there a reference study for this point?*

**Response:**
This steric hinderance for nitrate ionization was mentioned on the 6th paragraph of Section 3.2 in the Supplement of Berndt et al (2016). In the revised manuscript, although we have cited this paper in the same sentence, we have further clarified this as follows in line 503:
"Second, although underestimation of certain HOM $RO_2$ formed from α-pinene+OH reaction has been reported (Berndt et al., 2016), such underestimation was mainly attributed to the steric hinderance in forming HOM-nitrate cluster for HOM with bicyclic structures ($C_{10}H_{17}O_7•$) ((Berndt et al., 2016), Section 3.2 of the Supplement therein) and thus such underestimation is not common for all HOM."

*Why is there a steric hinderance for nitrate ionization, but not acetate ionization?*
**Response:**
As mentioned on 7th paragraph of Section 3.2 in the Supplement of Berndt et al (2016), the original interpretation of the reason is as follows: "The $RO_2 \cdot CH_3COO^-$ clusters are all more strongly bound than the corresponding $RO_2 \cdot NO_3^-$ clusters, both in an absolute sense, and relative to the neutral acid-ion cluster. Even the presence of a single peroxide or carboxylic acid group is enough to make the binding of a $RO_2$ radical to acetate competitive with that of acetic acid. This explains why acetate CIMS is highly effective at detecting products of both OH- and $O_3$-initiated autoxidation. The binding of $RO_2$ radicals with two H-bond donating functional groups to $CH_3COO^-$ is more than 10 kcal/mol stronger than the binding of acetic acid to $CH_3COO^-$. Thus, acetic acid is not able to compete with the multiply substituted $RO_2$ at any reasonable concentration ratio, explaining the lack of dependence of the detection efficiency of autoxidation products on the acetic acid concentration, see results in Fig. 3. As expected, the relative sensitivity of acetate CIMS to carboxylic acid groups compared to OH or OOH groups is also much

larger than that of nitrate CIMS. If OH – initiated autoxidation has a larger probability of forming carboxylic acid groups than $O_3$ – initiated autoxidation, this may also help explain the differences in relative sensitivities toward the two groups of products.".

*More importantly, the authors stated that bicyclic structures are not common for all HOMs. But in Schemes 1 and 2 in this manuscript, I see a lot of bicyclic structures.*

**Response:**

We apologized for the ambiguity of this sentence. In fact, we would like to state that underestimation attributed to the steric hinderance in forming HOM-nitrate cluster for HOM with bicyclic structures ($C_{10}H_{17}O_7\bullet$) is not common. We revised this sentence in line 505 in revised main text:

"…with bicyclic structures ($C_{10}H_{17}O_7\bullet$) ((Berndt et al., 2016), Section 3.2 of the Supplement therein) and thus such underestimation is not common for all HOM."

Although some HOM contain bicyclic structures in Scheme 1 or Scheme 2, most bicyclic products contain less than 6 oxygen atoms. They are not HOM $RO_2$, and as early generation $RO_2$, they can undergo further autoxidation to form HOM. Also, even if $C_{10}H_{17}O_7\bullet$ in Scheme 1 has bicyclic structures, the $C_{10}H_{17}O_7\bullet$ $RO_2$ are less intensity and not dominant in total $C_{10}H_{17}O_x$ family as shown in Fig. S9.

Overall, we agree that some HOM from both α-pinene ozonolysis and α-pinene + OH oxidation have different sensitivities in $NO_3^-$-CIMS as reported by Berndt et al. (2016). However, the difference was mostly attributed to bicyclic structures of some HOM e.g. $C_{10}H_{17}O_7\bullet$ formed in OH oxidation. Such difference is not applicable to HOM from OH-addition and HOM from OH H-abstraction. It is reasonable to expect a generally similar sensitivity for HOM in this study by given three reasons in our last response.

In the revised manuscript, we have further added a note in the "HOM yield" section in line 562.

"We further note that these HOM yields may be subject to uncertainties due to the assumption that HOM have the same sensitivity as $H_2SO_4$ as we discussed in Sect. 3.3.3. As mentioned above, our previous study showed that using an unified sensitivity of $H_2SO_4$ only leads to a maximum uncertainty of a factor of two by comparing the condensation HOM and corresponding increase of aerosol mass (Pullinen et al., 2020)."

*Initial comment #2:*

*With the amended chemical reactions, it should be realized that the yields of different functional groups (ROH, R=O, and ROOH) are highly structure dependent. Carbonyl yield + alcohol yield does not necessarily equal to 1; the values of alpha and beta largely vary with structure; alkoxy radical fate also largely affects the yields. My whole point is, with such complexity, deriving these equations (Eq. 2-8) is not very meaningful.*

**Response:**

We agree that the yields of different functional groups (ROH, R=O, and ROOH) are highly structure dependent. As HOM in this study is likely formed via a 6-membered carbon ring opening as discussed below, most HOM $RO_2$ are likely primary or secondary $RO_2$ as shown in Scheme 1 and Scheme 2 and $RO_2$ distribution in Fig. S9. For primary and secondary HOM $RO_2$, although carbonyl yield and alcohol yield does not necessarily equal to 1, they are most likely to be 1 according to Jenkin et al (2019). With these equations, we can estimate carbonyl fractions formed via $C_{10}H_{17}O_x\bullet$ under reasonable assumption. At the same time, we have estimated the uncertainty of the value.

We added discussion in line 206 of the revised main text:

"As HOM in this study is likely formed via a 6-membered carbon ring opening as discussed below, most HOM $RO_2$ are likely primary or secondary $RO_2$ as shown in Scheme 1 and Scheme 2 and $RO_2$ distribution in Fig. S9. For primary and secondary HOM $RO_2$, although carbonyl yield and alcohol yield does not necessarily equal to 1, they are most likely to be 1 according to Jenkin et al (2019). With these equations, we can estimate carbonyl fractions formed via $C_{10}H_{17}O_x\bullet$ under reasonable assumption."

*Initial comment #4:*

*Part of the response is confusing. I thought that the authors suggested that the HOMs from the OH abstraction channel is through RO isomerization, not RO2. Then, "Y1 is the fraction of primary RO2 undergoing autoxidation to form highly oxygenated C10H15Ox" means that Y1 is nearly zero?*

**Response:**

Y1 refers to the fraction of primary $RO_2$ undergoing auto-oxidation to form highly oxygenated $C_{10}H_{15}O_x\bullet$, no matter whether they undergo direct auto-oxidation or via RO isomerization and further auto-oxidation. In the revised manuscript, we have added a note to avoid potential confusion as follows:

"$Y_1$ is fraction of primary $RO_2$ undergoing auto-oxidation to form highly oxygenated $C_{10}H_{15}O_x\bullet$, no matter whether they undergo direct auto-oxidation or via RO isomerization and further auto-oxidation."

*Considering the additional step to form RO from RO2+RO2, Eq, R1 is wrong. What are the ranges of kuni_1, kuni_2, and k[NO] anyway?*

**Response:**

In this study, the dominant consumption of $RO_2$ is mainly through $RO_2$ with NO channel (Fig. S3 in Supplement). Therefore, Eq. R1 is tenable even if considering the reactions of $RO_2+RO_2$ to form RO. Unfortunately, both $k_{nui\_1}$ and $k_{nui\_2}$ are unknown. That's why we assume them in the calculation. When $k_{nui\_1}$ is higher than $k_{nui\_2}$, one can get the ratio of $[C_{10}H_{15}O_x\bullet]$ to $[C_{10}H_{17}O_x\bullet]$ increases with increasing NO concentration. k[NO] are in the range 2 $s^{-1}$ to $3\times10^{-2}$ $s^{-1}$.

*"From this, we infer that the average reactive rate of C10H17Ox• at high NO is higher than that at low NO, which finally result in the increases of C10H17Ox• consumption. This inference is supported by the higher relative contribution of C10H17NOx at high NO (36.3%) than at low NO (16.1%) in Fig. 2."*
*Wouldn't this statement work the same way for C10H15Ox?*

**Response:**

We realize that this sentence did not provide extra support to the discussion, so we decide to delete this sentence. At the same time, we move the discussion of NO dependence of the original manuscript to Sect. 3.3.3 the line 436 of revised main text:

"The $NO_x$ dependence for $C_{10}H_{15}O_x\bullet/C_{10}H_{17}O_x\bullet$ may be attributed to the differences in their reactivity. One explanation for the $NO_x$ dependence is that the autooxidation of $C_{10}H_{15}O_x\bullet$ $RO_2$ radicals may be faster than that of $C_{10}H_{17}O_x\bullet$ $RO_2$ radicals, which leads to the lower concentration of $C_{10}H_{15}O_x\bullet$ at high NO and thus higher sensitivity to NO concentrations. Ratios of $C_{10}H_{16}O_x$(R-OH/OOH)/$C_{10}H_{15}O_x\bullet$ can be derived of 0.47 and 0.02 at low and high NO, respectively. The decrease of $C_{10}H_{16}O_x$(R-OH/OOH)/$C_{10}H_{15}O_x\bullet$ at high NO compared to low NO was more evident than the decrease of $C_{10}H_{18}O_x/C_{10}H_{17}O_x\bullet$. Theoretically, though, they should be similar. The difference may be attributed to the shift in $C_{10}H_{15}O_x\bullet$ distribution with different number of O, as evident in Fig. S6, and different isomers at high NO compared to low NO. At high NO there might thus be more $C_{10}H_{15}O_x\bullet$ that react slower with $HO_2$ or have a lower branching ratio forming ROOH in $RO_2+HO_2$, which depends on the explicit $RO_2$

structure, or have a lower yield forming ROH in $RO_2+RO_2$."

*Initial comment #5:*

*Thanks for the analysis regarding secondary chemistry. It appears that secondary chemistry has increased significantly after ~ 5 min. I wonder if the same analysis is performed comparing first 5-min vs. 15-min, what contributions would be for the OH-abstraction chemistry in limonene+OH oxidation. With the 15-min analysis, the authors suggest that C10H15Ox-derived HOMs contribute 41-42% of total C10-HOMs. Would this number be lower within the first 5 min?*

**Response:**

We guess that the reviewer refer to Fig. S9b, where the significant change at ~5 min at high NO (Fig. S9b) is attributed to the uncertainty of measured limonene consumption. Actually, as shown in Fig. S9, the importance of secondary chemistry within the first 15 min did not change significantly based on both the experiment data or MCM simulation.

We have clearly shown that secondary chemistry is not important in the first 15 min. At the first 5 min, the contribution of $C_{10}H_{15}O_x$-derived HOMs in total $C_{10}$-HOMs are 41.1% at low NO, and 48.3% at high NO. Note that there would be higher uncertainty in the very earlier stage of the experiments. Considering the uncertainty, the contribution of $C_{10}H_{15}O_x$-derived HOMs in total $C_{10}$-HOM in the first 5 min is similar to the contribution of the first 15 min.

The caption of the Fig. S9 has been revised as follows:

"The large change in panel (b) at ~5 min results from the large measurement uncertainty of low accumulated limonene consumption measured by PTR-ToF-MS in the first few minutes."